# Lyl-1 regulates primitive macrophages and microglia development

Shoutang Wang [1,4,8], Deshan Ren[1,5,8], Brahim Arkoun[1], Anna-Lila Kaushik[1,6], Gabriel Matherat [1,7], Yann Lécluse[2], Dominik Filipp[3], William Vainchenker [1], Hana Raslova [1], Isabelle Plo[1] & Isabelle Godin [1✉]

During ontogeny, macrophage populations emerge in the Yolk Sac (YS) via two distinct progenitor waves, prior to hematopoietic stem cell development. Macrophage progenitors from the primitive/"early EMP" and transient-definitive/"late EMP" waves both contribute to various resident primitive macrophage populations in the developing embryonic organs. Identifying factors that modulates early stages of macrophage progenitor development may lead to a better understanding of defective function of specific resident macrophage subsets. Here we show that YS primitive macrophage progenitors express Lyl-1, a bHLH transcription factor related to SCL/Tal-1. Transcriptomic analysis of YS macrophage progenitors indicate that primitive macrophage progenitors present at embryonic day 9 are clearly distinct from those present at later stages. Disruption of *Lyl-1* basic helix-loop-helix domain leads initially to an increased emergence of primitive macrophage progenitors, and later to their defective differentiation. These defects are associated with a disrupted expression of gene sets related to embryonic patterning and neurodevelopment. Lyl-1-deficiency also induce a reduced production of mature macrophages/microglia in the early brain, as well as a transient reduction of the microglia pool at midgestation and in the newborn. We thus identify Lyl-1 as a critical regulator of primitive macrophages and microglia development, which disruption may impair resident-macrophage function during organogenesis.

[1] Gustave Roussy, INSERM UMR1287, Université Paris-Saclay, Villejuif, France. [2] PFIC, IUMS AMMICa (US 23 INSERM/UMS 3655 CNRS; Gustave Roussy, Villejuif, France. [3] Laboratory of Immunobiology, Institute of Molecular Genetics of the Czech Academy of Sciences, Prague, Czech Republic. [4] Present address: Department of Pathology and Immunology, Washington University School of Medicine, St. Louis, MO 63110, USA. [5] Present address: Ministry of Education Key Laboratory of Model Animal for Disease study; Model Animal Research Center, Medical school of Nanjing University, Chemistry and Biomedicine Innovation center, Nanjing University, Nanjing 210093, China. [6] Present address: Plasseraud IP, 33064 Bordeaux, France. [7] Present address: Agence Nationale pour la Recherche, Paris, France. [8] These authors contributed equally: Shoutang Wang, Deshan Ren. ✉email: Isabelle.Godin@gustaveroussy.fr

mongst the components of the transcription factor network that regulate hematopoietic cells features, *Tal-1*, *Lmo2*, *Runx1*, and *Gata2* stand out as major regulators of hematopoietic progenitor development[1,2]. *Tal-1*, *Lmo2*, and *Gata-2* belong to a transcriptional complex, which also includes the basic helix–loop–helix (bHLH) transcription factor lymphoblastic leukemia-derived sequence 1 (*Lyl-1*). Unlike its paralog *Tal-1*, which is mandatory for the specification of all hematopoietic progenitors[3,4], Lyl-1 roles during developmental hematopoiesis remains poorly characterized. We analyzed these functions at the onset of YS hematopoietic development using *Lyl-1^{LacZ/LacZ}* mutant mice[5].

During ontogeny, hematopoietic progenitors are generated in three successive and overlapping waves[6,7]. The emergence of the Hematopoietic Stem Cells (HSC) that will maintain lifelong hematopoiesis in the adult occurs at mid-gestation in the third and definitive hematopoietic wave. HSC generated in the aorta region immediately migrate to the fetal liver (FL) where they mature and amplify before homing to the bone marrow before birth[8,9]. Prior to HSC generation, the production of blood cells relies on two hematopoietic waves provided by the YS. This HSC-independent hematopoiesis comprises first the primitive hematopoietic wave, with the transient production

of progenitors with embryonic specific features: From Embryonic day (E) 7.00, the YS produces monopotent progenitors for erythrocytes, megakaryocytes and macrophages (MΦ)[10,11], along with bipotent Erythro-Megakaryocytic progenitors[12], in a *Myb*-independent pathway[13,14]. The second YS wave, called transient-definitive, provides for a limited duration progenitors (mostly erythro-myeloid) that seed the FL and produce a hematopoietic progeny that displays definitive/adult differentiation features. Erythro-myeloid cell production in this wave occurs in a *Myb*-dependent pathway, through the progressive differentiation of erythro-myeloid progenitors (EMP) in a pathway similar to the adult one[6]. As primitive and transient definitive YS waves both produce cells from erythro-myeloid lineages, they are also termed respectively "early EMP" and "late EMP"[15,16].

Considering the MΦ lineage, fate-mapping approaches aimed at determining the embryonic origin of resident-MΦs indicated that most tissues harbor resident-MΦs of diverse origins (YS, FL and adult bone marrow)[15,17,18], which complicates the characterization of wave-dependent functions of the various subsets. However, these fate-mapping analyses established that, contrary to others tissue, brain MΦs (microglia and Border Associated MΦ (BAM)) develop only from YS-derived MΦ-progenitors[14,16,19–21], confirming a model we previously put forward[22]. Due to the

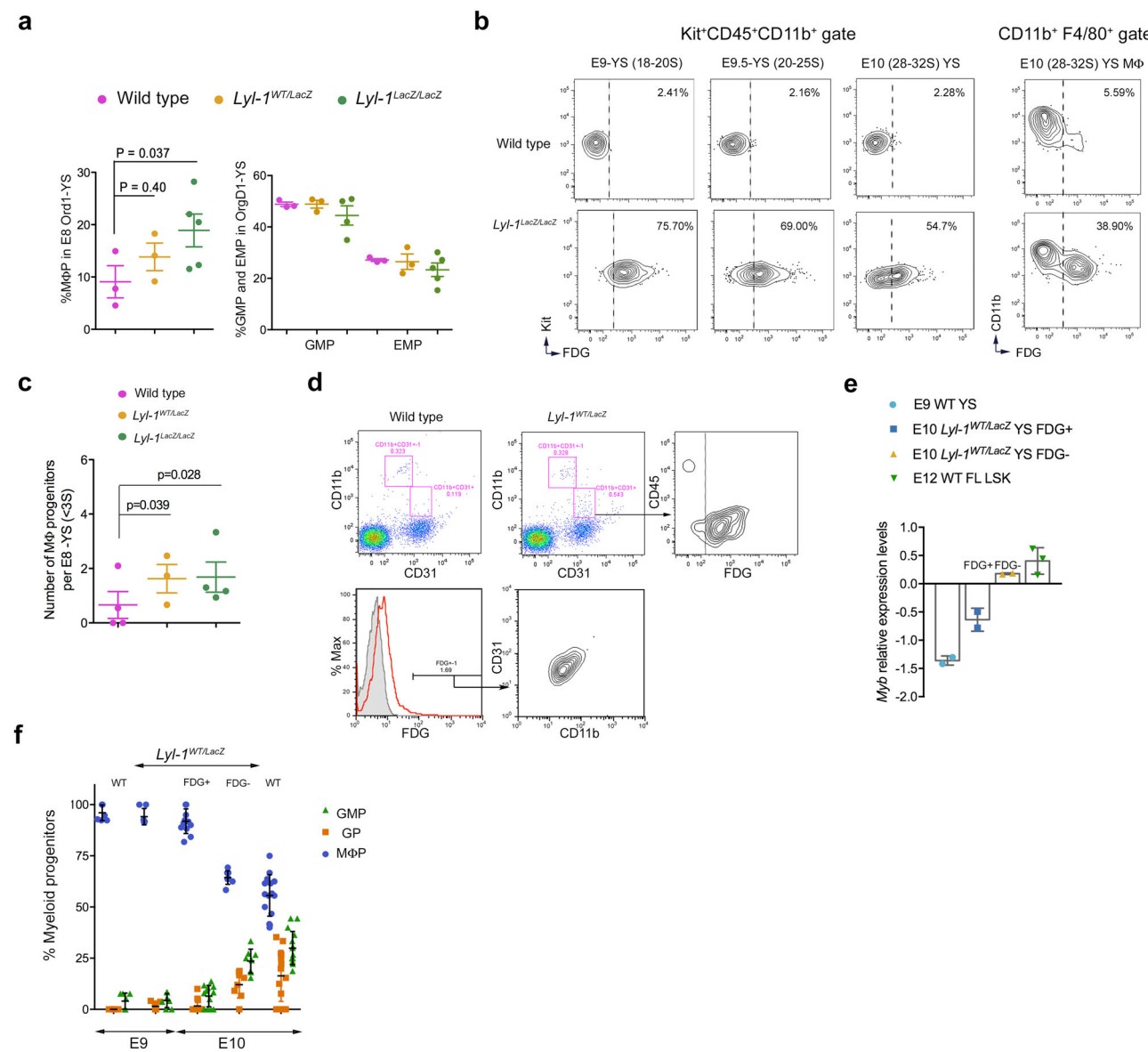

**Fig. 1 Lyl-1 expression marks MΦPrim progenitors in the early YS. a** Lyl-1-deficiency leads to an increased production of MΦ-progenitors in the early YS: Clonogenic potential of E8 OrgD1-YS cells: production of MΦ-progenitors (MΦP) in WT, *Lyl-1*$^{WT/LacZ}$ and *Lyl-1*$^{LacZ/LacZ}$ OrgD1-YS. Distribution of other progenitors with a myeloid potential (EMP and GM) in WT, *Lyl-1*$^{WT/LacZ}$ and *Lyl-1*$^{LacZ/LacZ}$ E8 OrgD1-YS. (n = 3–5, 3–6 YS per sample; mean ± s.e.m.; Unpaired, two-tailed *t*-test). **b** Lyl-1 expression in MΦ-progenitors: FACS-Gal assay, using the β-Gal fluorescent substrate FDG, was used as a reporter for Lyl-1 expression. While all MΦ-progenitors in E9-YS expressed FDG/Lyl-1, E9.5 and E10-YS harbored two MΦ-progenitor subsets discriminated by their FDG/Lyl-1 expression. FDG⁺/Lyl-1⁺ and FDG⁻/Lyl-1⁻ mature MΦs (CD11b⁺F4/80⁺) also coexisted in E10-YS. The contour plots in WT samples indicate the level of non-specific background β-Gal activity/FDG labeling in WT samples. Representative profiles of 3 independent samples, each consisting of 3–4 YS (see gating strategy in Supplementary Fig. 1a). **c** In clonogenic assays, the number of MΦ-progenitors obtained from E8-YS (0-3 Somites) was increased in *Lyl-1*$^{WT/LacZ}$ and *Lyl-1*$^{LacZ/LacZ}$ compared to WT. The majority of the 25–30 progenitors per YS were Ery$^P$ (60 to 80% in each 3 genotypes). Other progenitors were occasionally and randomly detected in WT and mutant samples (less than one EMP (0.81% ± 0.66; n = 3) and/or GM progenitor per E8-YS), confirming that the assay was performed at a time when MΦ$^{T-Def}$ progenitors were absent. (n = 3–4, 5–10 YS per sample; error bars show mean ± s.e.m.; Unpaired, two-tailed *t*-test). **d** MΦ$^{Prim}$ progenitors express Lyl-1. Flow cytometry profiles of WT and *Lyl-1*$^{WT/LacZ}$ E8-YS (0-3 S). CD11b⁺CD31⁻ MΦs correspond to maternal MΦs present in E8-YS[11]. All CD11b⁺CD31⁺ MΦ-progenitors displayed FDG/Lyl-1 expression (Red line). The contour plots in WT samples indicate the level of non-specific background β-Gal activity/FDG labeling in WT samples. Shown are representative profiles of 4 independent experiments. **e** RT-qPCR quantification of *Myb* expression levels: Kit⁺CD45⁺CD11b⁺ progenitors were sorted from WT E9-YS, WT and *Lyl-1*$^{WT/LacZ}$ E10-YS, and from FDG/Lyl-1 positive and negative fractions of MΦ-progenitors from *Lyl-1*$^{WT/LacZ}$ E10-YS. Lin⁻Sca⁺Kit⁺ (LSK) progenitors from WT E12-FL were used as positive control. FDG⁺/Lyl-1⁺ MΦ-progenitors from E10-YS expressed *Myb*$^{Low/Neg}$ levels similar to MΦ$^{Prim}$ progenitors from E9-YS. The FDG⁻/Lyl-1⁻ fraction expressed significantly higher *Myb* levels, similar to LSK cells from E12-FL. *Myb* expression levels, shown on a Log² scale, were normalized to the mean expression value obtained for WT E10-YS, considered as 1 (Each dot represent an independent experiment; Error bars show means ± s.e.m; Unpaired, two-tailed *t*-test). **f** FDG/Lyl-1 positive and negative myeloid progenitors produce a distinct progeny: Clonogenic assays characterization of the type of progenitors produced by myeloid progenitors (Ter119⁻Kit⁺CD45⁺CD11b⁺) sorted from WT and *Lyl-1*$^{WT/LacZ}$ E9-YS ( < 18 S; n = 7) and E10-YS (n = 15) in 3 independent experiments. At E10, myeloid progenitors from *Lyl-1*$^{WT/LacZ}$ YS were subdivided into FDG/Lyl-1 negative (n = 15) and positive (n = 12) fractions (5 independent experiments). Samples were biological replicates comprising 6–8 YS. 100 to 150 Kit⁺CD45⁺CD11b⁺ cells per condition were platted in triplicate. All samples produced few non-myeloid contaminants, such as Erythro-Megakaryocytic progenitors and EMP in similar, non-significant amounts. FDG⁺/Lyl-1⁺ progenitors essentially produced MΦ−progenitors (MΦP), while FDG⁻/Lyl-1⁻ progenitors produced also granulo-monocytic- (GMP) and granulocyte- (GP) progenitors. Error bar shows means ± s.e.m.

coexistence of two waves in the YS, the origin of microglia has been debated (reviewed in refs. [6,15,18,23]). An origin of microglia from MΦ-progenitors from the primitive/"early EMP" wave was supported by microglia labeling following an early (E7.0-E7.5) CRE-mediated induction of Runx1[19] and by the intact microglia pool in *Myb*-deficient mice[14,21]. The origin of microglia was also attributed to the primitive wave in zebrafish embryos, since primitive macrophage (MΦ$^{Prim}$) progenitors arise in this species from a location distinct from other hematopoietic progenitors[24–26]. Finally, the normal microglia development in mice lacking Kit ligand, leading to an impaired EMP development and the depletion of resident-MΦs in the skin, lung, and liver supports this model[27].

We here show that, at the early stages of YS hematopoiesis, *Lyl-1* expression characterizes primitive MΦ-progenitors. Through RNA-seq. analyses, it appears that these primitive MΦ-progenitors harbor an immune-modulatory phenotype, while those produce at a later stage favor the inflammatory signaling which promotes the emergence of HSC in the third and definitive hematopoietic wave[28].

Our results also indicate that in the brain, Lyl-1 is expressed in the entire microglia/BAM cell population at the onset of brain colonization and appeared to regulate microglia/BAM development.

Altogether, these data point to Lyl-1 as a major regulator of early embryonic MΦ-progenitors development and advocate for further analyses to more precisely delineate Lyl-1 function during the development of resident-MΦs in homeostatic and pathological contexts.

## Results and discussion

**Lyl-1 expression marks MΦ$^{Prim}$ progenitors from the early YS.** *Lyl-1* being expressed in the YS from the onset of YS hematopoiesis[29], we first explored its function by characterizing the clonogenic potential of WT, *Lyl-1*$^{WT/LacZ}$ and *Lyl-1*$^{LacZ/LacZ}$ YS. E8-YS were maintained in organ culture for 1 day (E8 OrgD1-YS), allowing only the development of primitive and

transient-definitive progenitors[30,31]. Compared to WT, the production of MΦ progenitors was increased in *Lyl-1*$^{WT/LacZ}$ and *Lyl-1*$^{LacZ/LacZ}$ OrgD1-YS. Otherwise, the clonogenic potential and progenitor distribution were unmodified (Fig. 1a).

Using FACS-Gal assay, we noticed that the entire MΦ-progenitor population (Kit⁺CD45⁺CD11b⁺) expressed Lyl-1 at E9. In contrast, two MΦ-progenitor subsets, discriminated by FDG/Lyl-1 expression, were present after E9.5 (Fig. 1b). Since after E9.5, the YS harbors both MΦ$^{Prim}$ and transient-definitive (MΦ$^{T-Def}$) MΦ−progenitors, and as these progenitor subsets cannot be discriminated by phenotype[11], we investigated the known features discriminating these two waves: the origin from monopotent progenitors for MΦ$^{Prim}$ progenitors[10,11] and the *Myb*-dependent[14,32] differentiation of MΦ$^{T-Def}$ progenitors from EMP, via the production of granulo-monocytic (GM-), then granulocyte (G-) and MΦ-progenitors[6].

At E8 (0-5 somites), when the YS only harbors MΦ$^{Prim}$ progenitors, clonogenic assays also pointed to an increased production of MΦ-progenitors in mutant E8-YS compared to WT (Fig. 1c). At this stage, all MΦ-progenitors, which harbor a CD11b⁺CD31⁺ phenotype[11], expressed FDG/Lyl-1 (Fig. 1d). Most FDG⁺/Lyl-1⁺ CD11b⁺CD31⁺ cells (69.27%±0.33%) from *Lyl-1*$^{WT/LacZ}$ E8-YS reliably produced MΦ colonies (72.78 ± 9.65%; n = 3) in clonogenic assays, amounting 1-4 MΦ progenitors per YS, a value consistent with previously published data[10,11].

Lyl-1 expression by MΦ$^{Prim}$ progenitors was strengthened by RT-qPCR comparison of *Myb* expression: Both E9-YS MΦ$^{Prim}$ progenitors and FDG⁺/Lyl-1⁺ progenitors from E10-YS expressed low *Myb* levels strengthening their primitive status, while FDG⁻/Lyl-1⁻ progenitors from E10-YS progenitors displayed *Myb* levels similar to lineage-negative Sca1⁺Kit⁺ progenitors from E12-FL (Fig. 1e).

The differentiation potential of FDG⁺/Lyl-1⁺ and FDG⁻/Lyl-1⁻ fractions of Ter119⁻Kit⁺CD45⁺CD11b⁺ myeloid progenitors isolated from E10-YS also pointed to Lyl-1 expression by MΦ$^{Prim}$ progenitors (Fig. 1f): Similar to WT E9-YS MΦ$^{Prim}$ progenitors, E10 FDG⁺/Lyl-1⁺ progenitors appeared monopotent, as they nearly exclusively produced MΦ colonies. In contrast, E10

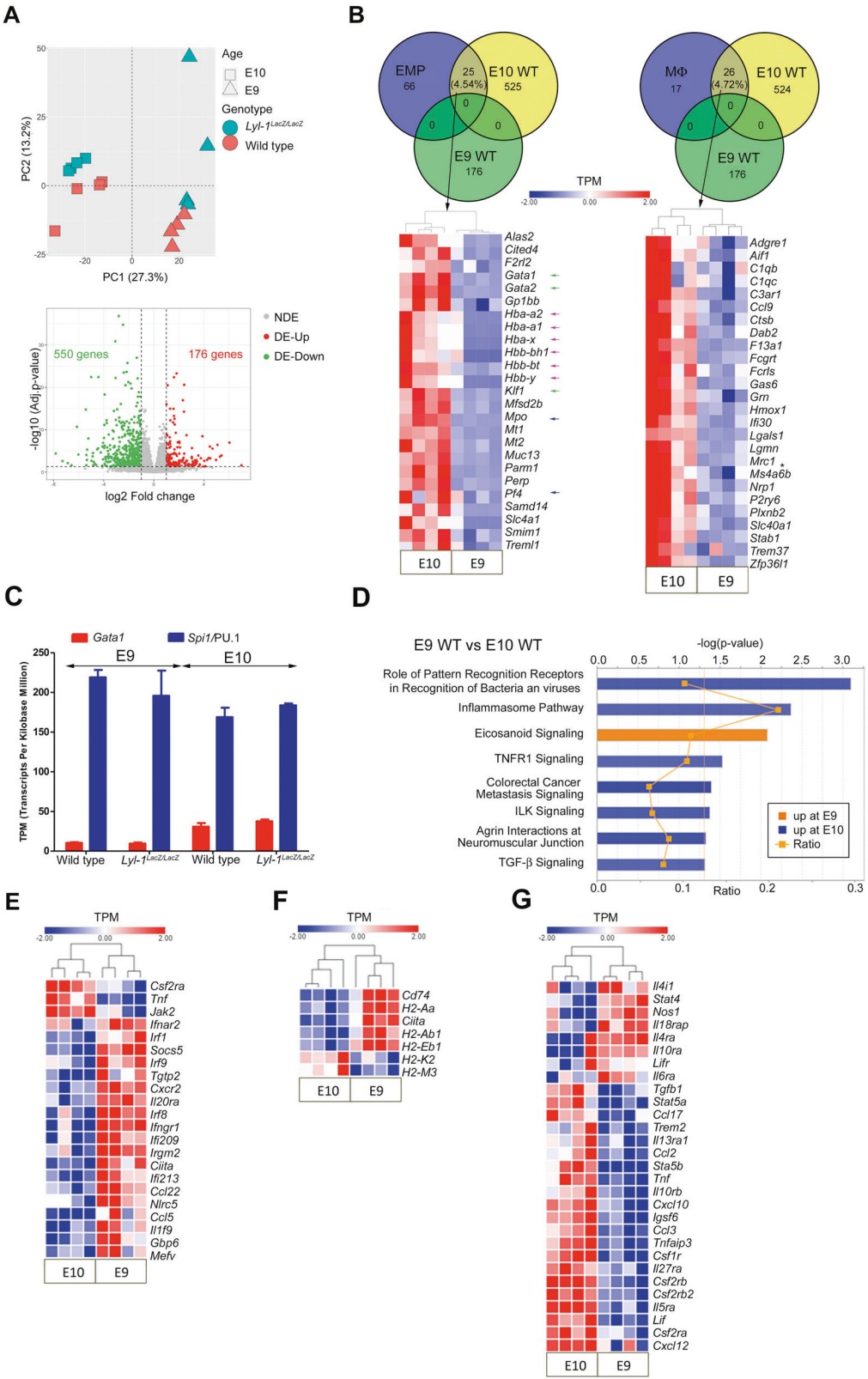

FDG⁻/Lyl-1⁻ myeloid progenitors produced GM, G and MΦ colonies, a feature typical of transient-definitive progenitors[6]. Overall, these data together suggested that Lyl-1 may mark MΦ^Prim progenitors from the earliest wave.

**Distinct features of WT MΦ-progenitors at E9 and E10.** The distinction between E9 and E10 MΦ-progenitors was confirmed

in RNA-seq. analysis of CD45⁺CD11b⁺Kit⁺ MΦ-progenitors sorted at E9 (MΦ^Prim progenitors) and E10 (MΦ^Prim and MΦ^T-Def progenitors). Principal Component Analysis separated E9 and E10 MΦ-progenitors according to stage and genotypes (Fig. 2A). E9 and E10 WT MΦ-progenitors differed by the expression of 726 genes, 176 being up-regulated at E9 and 550 at E10. Considering the coexistence of MΦ^Prim and MΦ^T-Def

**Fig. 2 Distinct features of WT MΦ-progenitors at E9 and E10. A** Differentially expressed genes in MΦ-progenitors (Kit[+]CD45[+]CD11b[+]) sorted from WT and *Lyl-1*[LacZ/LacZ] YS at E9 and E10. Unsupervised principal component analysis plot positioned E9 and E10 MΦ-progenitors in two distinct groups, followed by segregation of WT and *Lyl-1*[LacZ/LacZ] samples. Volcano plot of E9 WT vs E10 WT MΦ-progenitors: Red and green dots indicate genes with statistically significant changes in expression level. (*p*-value < 0.05, absolute fold change≥2.0) (NDE: not deregulated genes; DE-Up: up-regulated genes; DE-Down: down-regulated genes). **B** Venn diagram comparing differentially expressed genes in E9 WT vs E10 WT MΦ-progenitors to the EMP or MΦ signatures defined by Mass et al.[33] (GEO accession number GSE81774). The number and percentage of differentially expressed genes common to the EMP or MΦ signatures is shown. Expression profiles of the overlapping genes identified in the Venn diagram (Heatmap displays transformed log2-expression values; Unpaired *t*-test, two-tailed). Note the higher expression at E10 of genes involved in erythroid (Globins: Pink arrow; Transcription factors: green arrow), and megakaryocytic and granulocytic-related genes (blue arrow), and of *Mrc1*/CD206 (Asterisk). **C** Relative expression levels of *Gata1* and *Spi1*/PU.1, indicated by their relative Transcripts per million kilo-bases. **D** Enriched pathways in E9 and E10 WT MΦ-progenitors with absolute z-score ≥2, from QIAGEN's Ingenuity® Pathway Analysis. Bars: minus log of the *p*-value of each canonical pathway; Orange line: threshold *p*-value of 0.05. Ratio: genes detected/genes per pathway. **E** Expression profiles of differentially expressed genes related to IFNγ and IFNβ response, identified by g:Profiler (heatmap displays transformed log2-expression values; unpaired *t*-Test, two-tailed). **F** Expression profiles of differentially expressed genes related to MHC-II complex (Heatmap displays transformed log2-expression values; unpaired *t*-Test, two-tailed). **G** Expression profiles of differentially expressed genes related to cytokine signaling (Heatmap displays transformed log2-expression values; unpaired *t*-Test, two-tailed).

progenitors in E10-YS, differentially expressed genes found at E10 may reflect wave-specific differences or stage-dependent changes related to MΦ[Prim] progenitor maturation.

Overlapping the identified differentially expressed genes to the EMP and E10.25-E10.5 MΦs signatures obtained by Mass et al.[33] confirmed that WT E9 MΦ[Prim] progenitors were distinct from these two populations, since none of the 176 genes upregulated at E9 belonged to these signatures. Comparatively, about 5% of the genes up-regulated at E10 belonged to the EMP and MΦ signatures (Fig. 2B). A similar separation was observed in GSEA analyses (Supplementary Fig. 1b). These observations suggest that within E10 MΦ-progenitors some, likely the MΦ[T-Def] ones, retain part of the EMPs signature.

In a WT context, E9 MΦ[Prim] progenitors differed from E10 MΦ-progenitors by their transcription factors repertoire. Genes regulating erythroid development (*Gata1*, *Gata2*, *Klf1*) and globin genes, embryonic (*Hbb-bh1*, *Hba-x*, *Hbb-y*) and definitive (*Hba-a2*, *Hba-a1*, *Hbb-bt*), were enriched at E10 (Fig. 2B), while *Spi1*/PU.1 was highly expressed compared to *Gata1* at both stages (Fig. 2C). The lower expression level at E9 of erythroid genes and of genes involved in granulo-monocytic (*Mpo*, *Csf2r*/GM-CSF receptors, *Cebp*, *Jun*) and megakaryocytic development (*Pf4*, *TPO signaling*) (Fig. 2B; Supplementary Table 1) sustains the monopotent/primitive status of E9 MΦ-progenitors, and suggests that MΦ[T-Def] progenitors may retain the expression of genes that characterize their EMP ancestor.

Due to the simultaneous presence of MΦ[Prim] and MΦ[T-Def] progenitors in E10-YS, the differential expressions of markers that are wave-specific at these stages (Primitive: *Lyl-1*; transient-definitive: *Myb* and *Tlr2*[34]) were not significant despite a tendency to decrease for *Lyl-1* and increase for *Myb* and *Tlr2* (Supplementary Fig. 1c).

IPA and GSEA analyses indicated that E9 MΦ[Prim] progenitors were more active in Eicosanoid signaling than E10 progenitors (Fig. 2D). They were also enriched in type I interferon (IFN) β and type II IFNγ signaling (Fig. 2E) and in MHC-II related genes, especially *Cd74* (top 1 IPA network) (Fig. 2F; Supplementary Fig. 1d). Cytometry analyses confirmed a low, but significant, enrichment of MHC-II expression at E9, compared to E10 (Supplementary Fig. 1e). Comparatively, E10 MΦ-progenitors were more active in inflammatory signaling (Fig. 2D, E; Supplementary Fig. 1d, f–h; Supplementary Table 1), and metabolically active (Supplementary Table 1). The complement cascade and phagocytosis also prevailed at E10 (Supplementary Fig. 1i, j).

Altogether, the signature for WT E9 MΦ[Prim] progenitors points to an immuno-modulatory function, while E10 MΦ-progenitors appear involved in phagocytosis and inflammatory signaling. Interestingly, inflammatory signaling has been revealed

as a key factor favoring embryonic HSC emergence (reviewed in ref. [35]). The source of inflammatory signals was further identified as MΦ-progenitors expressing *Mrc1*/CD206[36], a marker up-regulated in E10 WT MΦ-progenitors compared to E9 (Fig. 2B).

**Lyl-1 deficiency impacts embryonic development.** When evaluating the effect of *Lyl-1*-deficiency at the earliest stage of MΦ[Prim] development, clonogenic assays pointed to an increased production of MΦ-progenitors in *Lyl-1*[LacZ/LacZ] MΦ[Prim] E8-YS compared to WT (Fig. 1c). The analysis of the RNA-seq. comparison of E9 WT and *Lyl-1*[LacZ/LacZ] MΦ[Prim] suggests that the increased size of the initial MΦ-progenitor pool could results from an elevated commitment of mesodermal/pre-hematopoietic cells to a MΦ fate, rather than from a defective differentiation (Supplementary Fig. 2a–d and Supplementary note 1). The high increase of *Itga2b*/CD41 expression level in E9 *Lyl-1*[LacZ/LacZ] MΦ[Prim] progenitors (Fig. 3a) may reflect this elevated commitment. Lyl-1 expression in YS mesoderm, where it cannot substitute for *Tal-1* mandatory function for the generation of hematopoietic progenitors[4], was already established[3,29]. Recently, Lyl-1 was identified as a regulator of mesoderm cell fate[37] and of the maintenance of primitive erythroid progenitors[38].

The transcription factors network that controls developmental hematopoiesis[2] was also modified (Fig. 3b): beside the expected reduction of *Lyl-1* expression, the expression of *Lmo2*, a *Lyl-1* target[39], was down-regulated, while *Tal-1* up-regulation might reflect some compensatory function[3]. The consequences were apparent in GSEA analyses: both pathways and GO terms uncovered an up-regulation of signaling pathways involved in embryo patterning (Wnt, Hox and Smad) in *Lyl-1*[LacZ/LacZ] MΦ-progenitors, as well as a highly modified collagen, integrin, and cadherin usage (Supplementary Table 2). Accordingly, developmental trajectories were affected (Fig. 3c), with the up-regulation in E9 *Lyl-1*[LacZ/LacZ] MΦ-progenitors of gene sets related to "anterior-posterior pattern specification" and "anatomical structure formation involved in morphogenesis", notably skeletal and nervous system development.

GSEA and KEGG comparison of *Lyl-1*[LacZ/LacZ] and WT MΦ-progenitors at E10 highlighted another patterning modification, namely the down-regulation of gene sets involved in heart development (Supplementary Fig. 2e; Supplementary Table 3), which might stem from a defective MΦ development. The heart harbors three resident-MΦ subsets, two of which originate from the YS[40]. Amongst the features that distinguish WT E9 MΦ[Prim] progenitors from E10 MΦ-progenitors, the enriched expression of MHC-II (Fig. 2F) and poor expression of phagocytosis-related genes (Supplementary Fig. 1i) at E9 also characterize one of the two YS-derived CCR2[-] resident-MΦ subsets in the heart[40,41].

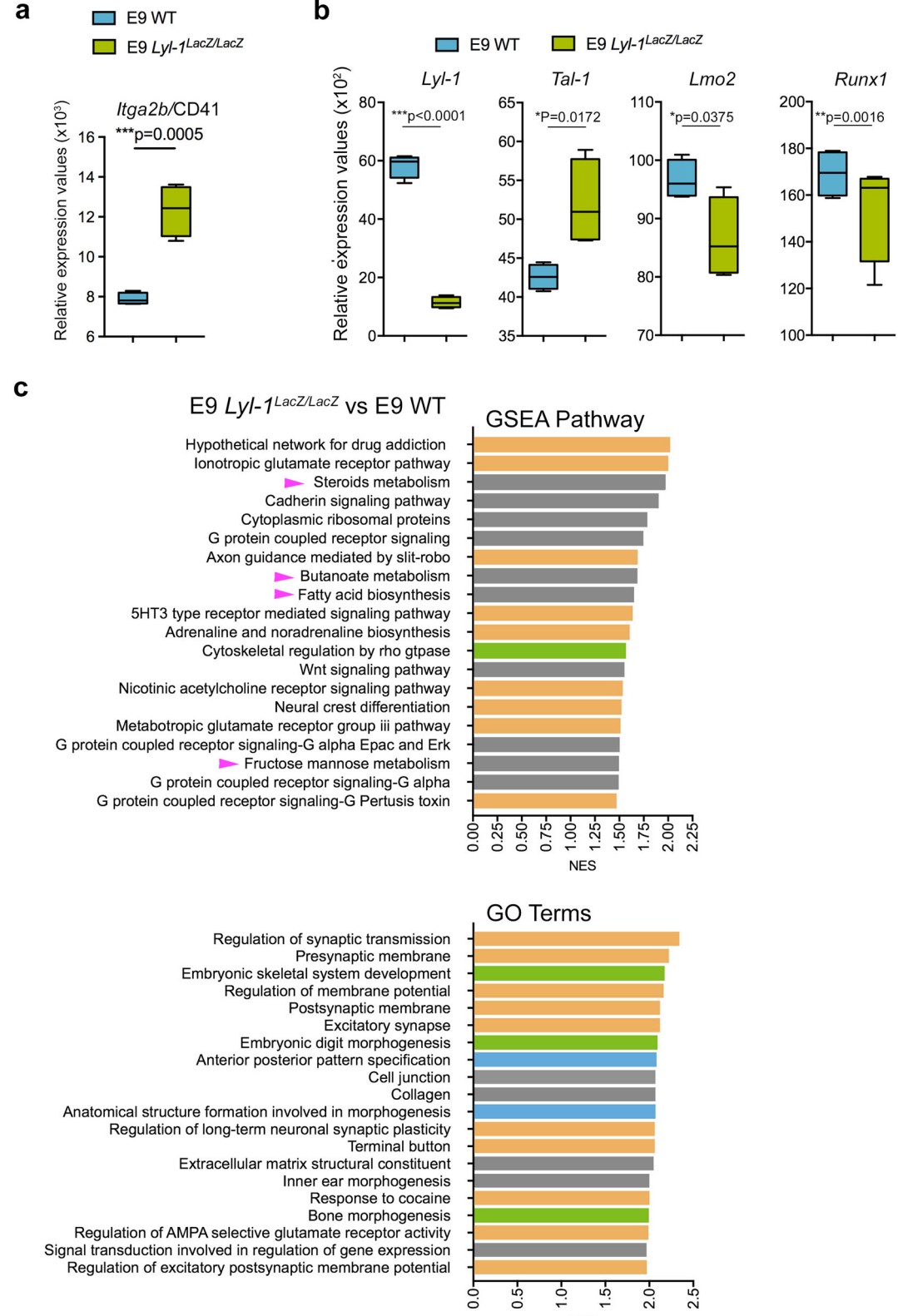

**Fig. 3 Lyl-1 regulates the production of E8 MΦPrim progenitors. a** Relative expression levels of the CD41 coding gene *Itg2b* in WT and *Lyl-1LacZ/LacZ* MΦ-progenitors at E9 (error bars show mean ± s.e.m.; unpaired *t*-Test, two-tailed). **b** Relative expression levels of transcription factors regulating hematopoietic progenitor emergence in *Lyl-1LacZ/LacZ* MΦ-progenitors compared to WT at E9 (Error bars show mean ± s.e.m.; unpaired, two-tailed *t*-Test). **c** GSEA pathways (FDR *q*-value <0.29) and GO terms (FDR *q*-value < 0.01) enriched in E9 *Lyl-1LacZ/LacZ* compared to E9 WT MΦ-progenitors. Highlighted are the pathways specifically related to embryo patterning (blue) and to the development of skeletal (green) and nervous systems (yellow). Pink arrows point to changes related to metabolic pathways.

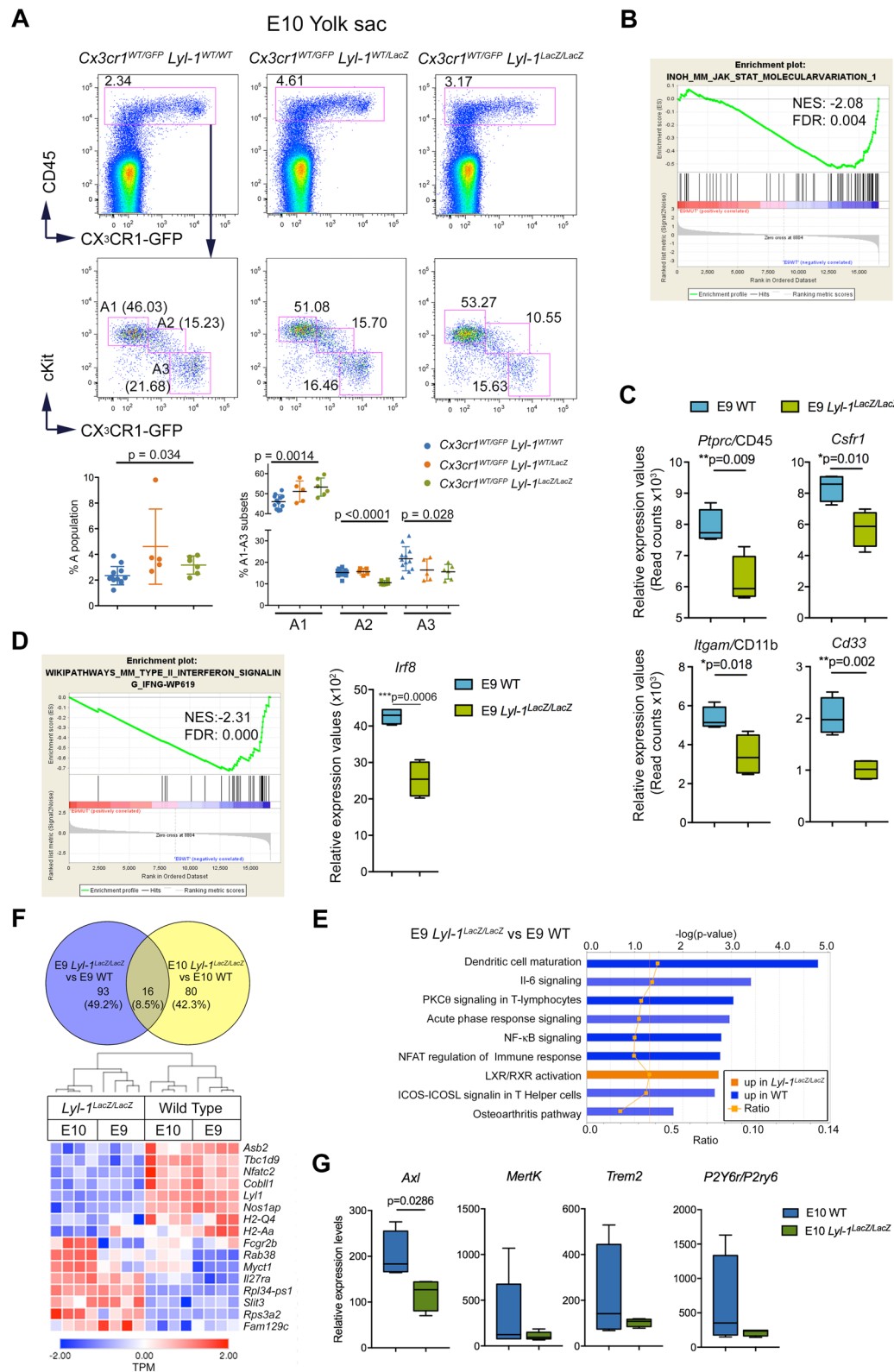

Therefore, a function for Lyl-1$^+$ MΦ$^{Prim}$ progenitors in heart development may be considered. This observation reinforces the need to better characterize the contributions of MΦ-progenitors from both primitive and transient-definitive waves to tissues harboring YS-derived resident-MΦs.

The patterning defects highlighted in defective MΦ$^{Prim}$, might be responsible, at least in part, for the significantly decreased litter

size and increased perinatal lethality observed in Lyl-1$^{LacZ/LacZ}$ mice compared to WT (Supplementary Fig. 2F), which indicates the requirement for Lyl-1 during various developmental processes. Whether this function is specific to mesodermal stage or related to a patterning function for MΦ$^{Prim}$ progenitors remains to be determined. Other Lyl-1-expressing lineages[29,42] might also participate to impaired development processes in Lyl-1$^{LacZ/LacZ}$

**Fig. 4 Defective differentiation of *Lyl-1^LacZ/LacZ^* MΦ-progenitors from E10-YS. A** Distribution of A1-A2 and A3 MΦ subsets in E10-YS from *Cx3cr1^WT/GFP^*: *Lyl-1^WT/WT^*, *Cx3cr1^WT/GFP^:Lyl-1^WT/LacZ^* and *Cx3cr1^WT/GFP^: Lyl-1^LacZ/LacZ^* embryos. While the size of the whole MΦ population is similar in the three genotypes, *Lyl-1*-deficiency leads to a modified distribution of the MΦ subsets with an increased size of the A1 subset and a reduced A3 pool (5–12 independent analyses, 6-8 YS per sample; Plots show mean ± s.e.m.; Unpaired, two-tailed *t*-Test). **B** GSEA pathway indicates a deficit in Jak1-Stat signaling in *Lyl-1^LacZ/LacZ^* MΦ-progenitors compared to WT at E9 (NES: normalized enrichment score; FDR: false discovery rate). **C** Relative expression levels (read counts) of hematopoietic markers in WT and *Lyl-1^LacZ/LacZ^* MΦ-progenitors at E9 (Error bars show mean ± s.e.m.; unpaired *t*-Test, two-tailed). **D** Top 1 GSEA pathway indicates that the IFN signaling pathway, which characterize E9 MΦ^Prim^ progenitors, is defective in *Lyl-1^LacZ/LacZ^* MΦ-progenitors (NES: normalized enrichment score; FDR: false discovery rate). The expression of *Irf8* is particularly affected (error bars show mean ± s.e.m.; unpaired *t*-test, two-tailed). **E** From the 53 canonical pathways identified by Ingenuity® Pathway Analysis as differentially expressed, 9 were enriched with an absolute Z score ≥ 1. Bars: minus log of the *p*-value of each canonical pathway; Orange line: threshold p-value of 0.05. Ratio: genes detected/genes per pathway. **F** Venn diagram comparing the differentially expressed genes in E9 *Lyl-1^LacZ/LacZ^* vs E9 WT to those in E10 *Lyl-1^LacZ/LacZ^* vs E10 WT MΦ-progenitors. Expression profiles of the differentially expressed genes common to both stages identified by the Venn comparison (Heatmap displays transformed log2-expression values; unpaired *t*-Test, two-tailed). **G** Relative expression levels (read counts) of phagocytosis genes in WT and *Lyl-1^LacZ/LacZ^* MΦ-progenitors at E10 (Box and whisker plots, min to max; median is shown; two tailed, unpaired Mann–Whitney test).

---

embryos, calling for the development of new mouse models that will allow lineage specific reports of Lyl-1 expression and function in endothelial or hematopoietic lineages during discrete steps of early YS development and beyond.

**Defective MΦ^Prim^ development in *Lyl-1* mutant YS.** The analysis of *Lyl-1* expression in A1-A2-A3 subsets from *Cx3cr1^WT/GFP^* YS indicated that *Lyl-1* is expressed throughout MΦ-progenitor differentiation, with levels decreasing from A1 to A3 (Supplementary Fig. 3a). We monitored the distribution of A1-A2 and A3 MΦ subsets (Supplementary Fig. 2a) at E10-YS, when all three subsets are present, using the *Cx3cr1^WT/GFP^:Lyl-1^LacZ/LacZ^* strain. While the size of the whole MΦ population was not overtly modified, *Lyl-1*-deficiency impacted the subset distribution, with increased A1 and reduced A2 and A3 pool sizes (Fig. 4A). Lyl-1 thus appears to regulate MΦ differentiation towards mature MΦs. A limited/delayed differentiation of E9 *Lyl-1^LacZ/LacZ^* MΦ^Prim^ progenitors could result from an altered cytokine signaling: Signaling through Jak-Stat pathway (*Jak1-Stat1/6*), which underlays cytokine signaling in hematopoietic cells, was down-regulated In E9 mutant progenitors, (Fig. 4B; Supplementary Fig. 3b), with profoundly decreased expression of interleukins and their receptors (*Il1a-b/Il1r1-2; Il4i1/Il4Ra; Il6st/Il6ra; Il10/ Il10ra*). It was also supported by a down-regulated *Spi1*/PU.1 signaling pathway (Supplementary Fig. 3c; Supplementary Tables 4–5) and by the decreased expression of *Ptprc*/CD45, *Csf1r, Itgam*/CD11b and CD33 (Fig. 4C).

Within the MΦ lineage, Lyl-1 function during normal development would initially consist to restrict the size of the MΦ^Prim^ progenitor pool and/or the duration of its production, which is transient[6], as indicated by the maintenance of the intermediate mesoderm to MΦ-progenitor pool observed in *Lyl-1^LacZ/LacZ^* E8-YS. Indeed, the increased size of the MΦ-progenitor pool in E8-E9 YS appears independent from the defective/delayed differentiation of MΦ-progenitors observed at E10, since this process starts after E9.5[10,11,15]. Subsequently, the increased size of MΦ^Prim^ progenitor pool in E10 *Lyl-1^LacZ/LacZ^* YS likely results from a defective/delayed differentiation mediated by a defective cytokine signaling, implying that during normal development, Lyl-1 would promote the differentiation of MΦ^Prim^ progenitors.

*Lyl-1^LacZ/LacZ^* MΦ-progenitors were also deficient in the IFN signaling that characterize E9 MΦ^Prim^ progenitors, notably *Irf8*, a factor involved in YS-MΦ and microglia development[21,43] (Fig. 4D). Compared to WT, MΦ-progenitors from *Lyl-1^LacZ/LacZ^* E9-YS up-regulated the LXR/RXR activation pathway (Fig. 4E) and metabolic pathways, some enriched WT MΦ-progenitors at E10 (Butanoate, steroid) (Supplementary Table 1), and other not (Fructose/mannose, fatty acid) (Fig. 3c). They were

also less active in inflammatory signaling pathways, particularly through NFκb, a factor known to interact with *Lyl-1*[44], and in TLR signaling (Fig. 4D; Supplementary Fig. 3b, d–e; Supplementary Table 5).

Overlapping the differentially expressed genes identified in *Lyl-1^LacZ/LacZ^* MΦ-progenitors at E9 and E10 identified the core signature of *Lyl-1*-deficiency, independent of the maturation occurring between these stages (Fig. 4F). Unfortunately, the co-existence of MΦ^Prim^ and MΦ^T-Def^ progenitors in E10-YS complicates the attribution of gene expression changes to a stage-dependent maturation of MΦ^Prim^ progenitors or to a signature specific to MΦ^T-Def^ progenitors. However, most pathways favored by E10 progenitors were insensitive to *Lyl-1*-deficiency, except for a decreased expression of phagocytosis genes (*Axl1, Mertk, Trem2, P2y6r/P2ry6*) (Fig. 4G). Conversely, the TLR signaling pathway was increased in *Lyl-1^LacZ/LacZ^* MΦ-progenitors at E10, compared to E9.

**Lyl-1-expressing MΦ-progenitors contribute to the fetal liver and brain.** The FL[9] and brain[19,22] are colonized as early as E9 by YS-derived resident-MΦ progenitors. We evaluated the contribution of Lyl-1-expressing MΦ^Prim^ progenitors to these rudiments at E10 (Fig. 5A). While E10-YS comprised FDG^+^/Lyl-1^+^ and FDG^-^/Lyl-1^-^ MΦ-progenitors and mature (F4/80^+^) MΦ subsets (Fig. 1b), the brain from the same embryos essentially harbored FDG^+^/Lyl-1^+^ MΦ-progenitors and MΦs. In contrast, both FDG^+^/Lyl-1^+^ and FDG^-^/Lyl-1^-^ MΦ-progenitors were present in E10-FL, as in E10-YS, and MΦ-progenitors were more abundant in mutant FL than in WT (Fig. 5B).

We next focused on brain MΦs during the colonization stage, which lasts until E11[45]. At this stage, microglia and perivascular, meningeal and choroid plexus MΦs, collectively referred to as BAMs, are all located in the brain mesenchyme and therefore undistinguishable[16,46]. FACS-Gal assay demonstrated that the whole F4/80^+^ microglia/BAM expressed Lyl-1 throughout the settlement period (Fig. 5C). The presence of FDG^+^/Lyl-1^+^F4/80^+^ microglia/BAM at early stage of brain colonization suggests that MΦs could participate to this step.

Lyl-1^+^ MΦ^Prim^ progenitors and early microglia/BAM shared similar features, such as an early appearance timing and low level of *Myb* expression (Fig. 5D), concordant with a *Myb*-independent development of microglia[14,21]. *Lyl-1* was also similarly expressed in A1-A2 and A3 MΦ subsets from the YS and brain (Fig. 5E; Supplementary Fig. 3a). *Lyl-1*-deficiency impacted the distribution of MΦs subsets in E10 *Cx3cr1^WT/GFP^:Lyl-1^LacZ/LacZ^* brain: an increased A1 and a reduced A3 pool size indicated that Lyl-1 regulates MΦ-progenitor differentiation in both YS and brain (Fig. 5F).

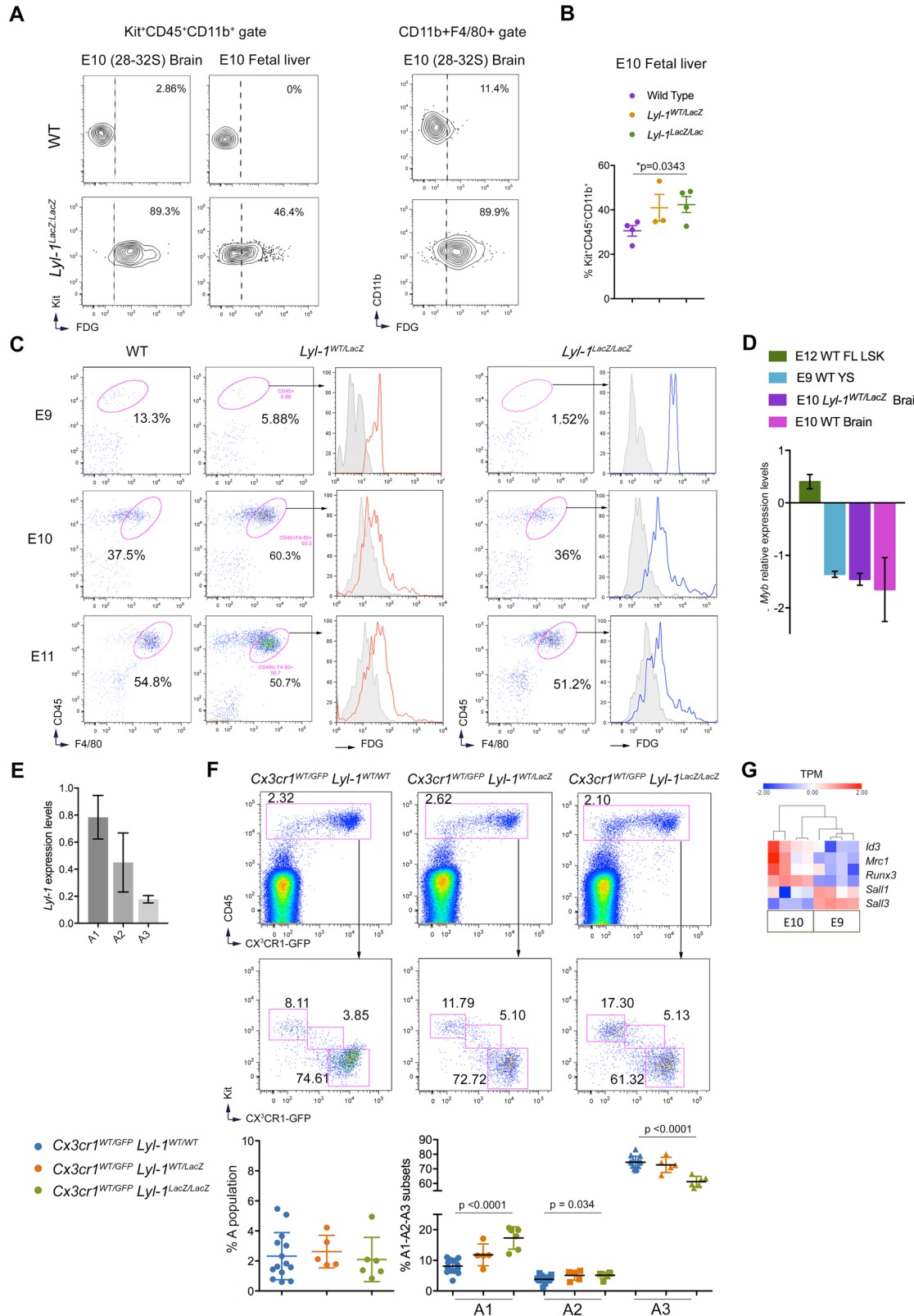

The proximity between YS MΦ$^{Prim}$ progenitors and microglia was also apparent in RNA-seq. data: E9 WT MΦ$^{Prim}$ progenitors expressed significantly lower *Mrc1*/CD206 and higher *Sall3* levels than E10 MΦ-progenitors, and a slightly increased *Sall1* level (Fig. 5G), a transcriptomic pattern that characterizes microglia[33,45,47]. This partial bias toward a microglia signature suggests that the first stage of microglia

development program is already initiated in MΦ$^{Prim}$ /"early EMP"progenitors in E9-YS.

**Lyl-1 inactivation impairs microglia development at two development stages.** Having defined Lyl-1 implication during microglia/BAM settlement in the brain, we turned to later development stages. Cytometry and database analyses[45]

**Fig. 5 Contribution of Lyl-1-expressing MΦ-progenitors to the fetal liver and brain. A** All MΦ-progenitors from E10-Brain expressed Lyl-1, contrary to the corresponding YS (Fig. 1b) which harbor both FDG$^+$/Lyl-1$^+$ and FDG$^-$/Lyl-1$^-$ subsets. MΦ-progenitors from E10 $Lyl$-$1^{LacZ/LacZ}$ FL harbored both FDG$^+$/Lyl-1$^+$ and FDG$^-$/Lyl-1 MΦ-progenitor subsets. in E10-brain, mature MΦs (CD11b$^+$F4/80$^+$ gate) were all FDG$^+$/Lyl-1$^+$. The contour plots in WT samples indicate the level of non-specific background β-Gal activity/FDG labeling in WT samples. Representative profiles of 3 independent samples, each consisting of 3-4 E10-Brain or 8-12 E10-FL. **B** Quantification of Kit$^+$CD45$^+$CD11b$^+$ MΦ-progenitors in E10-FL (plots show mean ± s.e.m.; Unpaired, two-tailed $t$-Test). **C** Lyl-1 marks the entire F4/80 + microglia/BAM population from the onset of brain colonization. FDG/Lyl-1 expression in F4/80+ microglia/BAM from the brain of $Lyl$-$1^{WT/LacZ}$ and $Lyl$-$1^{LacZ/LacZ}$ embryos at E9 to E11. The rare CD11b$^+$ F4/80$^{low-neg}$ cells present in the brain at E9 are FDG$^+$/Lyl-1$^+$. Gray histograms indicate non-specific background β-Gal activity/FDG levels in WT samples. Representative profiles of 3–5 independent samples, each consisting of 3-5 brains. **D** MΦ-progenitors from E10-brain express $Myb$ levels similar to E9-YS MΦ$^{Prim}$ progenitors. RT-qPCR quantification of $Myb$ expression levels in Kit$^+$CD45$^+$CD11b$^+$ MΦ-progenitors sorted from WT E9-YS and from WT and $Lyl$-$1^{WT/LacZ}$ brain at E10. Lin$^-$Sca$^+$cKit$^+$ (LSK) progenitors from WT E12-FL were used as positive control. $Myb$ expression levels, shown on a Log$^2$ scale, were normalized to the mean expression value obtained for WT E10-YS, considered as 1 (error bars show mean ± s.e.m.; Unpaired, two-tailed $t$-Test). **E** RT-qPCR analyses of $Lyl$-$1$ expression in A1 to A3 MΦ subsets isolated from $Cx3cr1^{WT/GFP}$ brain at E10. $Lyl$-$1$ is expressed by the 3 subsets, with levels decreasing with differentiation. Expression levels were normalized to the mean value obtained for $Cx3cr1^{WT/GFP}$ YS A1 progenitors ($n = 2$; Error bars show mean ± s.e.m.). **F** Defective differentiation of brain MΦ-progenitor in Lyl-1 mutant embryos. Distribution of A1-A2 and A3 MΦ subsets in E10 brain from $Cx3cr1^{WT/GFP}$:$Lyl$-$1^{WT/WT}$, $Cx3cr1^{WT/GFP}$:$Lyl$-$1^{WT/LacZ}$ and $Cx3cr1^{WT/GFP}$:$Lyl$-$1^{LacZ/LacZ}$ embryos. The size of the whole MΦ population was similar in the three genotypes, but $Lyl$-$1$ deficiency modified the distribution of the MΦ subsets with an increased size of the A1 subset and a reduced A3 pool (5-12 independent analyses, 6–8 brains per sample. Error bars show mean ± s.e.m.; Unpaired, two-tailed t Test). **G** Heatmap expression profile of the genes that mark the development of tissue resident-MΦs in WT E9 and E10 MΦ-progenitors (Heatmap displays transformed log2-expression values; Unpaired, two-tailed $t$-test).

---

confirmed the continuous expression of Lyl-1 in CD45$^{low}$ microglia until adulthood (Supplementary Fig. 4a). $LYL$-$1$ expression was also reported in microglia from healthy murine and human adults[48–51]. We examined the impact of $Lyl$-$1$-deficiency on microglia pool size during development. Microglia quantification pointed to E12 as the first step impacted. The arrested increase of microglia pool in $Lyl$-$1^{LacZ/LacZ}$ brain at E12 (Fig. 6a) resulted from a reduced proliferation (Fig. 6b) rather than an increased apoptosis (Supplementary Fig. 4b). Moreover, $Lyl$-$1$-deficiency provoked morphological changes in E12 $Cx3cr1^{WT/GFP}$:$Lyl$-$1^{LacZ/LacZ}$, compared to $Cx3cr1^{WT/GFP}$ microglia, namely a reduced number and extent of ramifications (Fig. 6c; Supplementary Fig. 4c, d). From E14, the microglia pool size returned to levels similar to WT (Fig. 6a), probably due to the highly reduced apoptosis level in $Lyl$-$1^{LacZ/LacZ}$ microglia at E14 (Supplementary Fig. 4b).

P0-P3 was identified as a second stage altered in $Lyl$-$1^{LacZ/LacZ}$ microglia. At birth, the cellularity of $Lyl$-$1^{LacZ/LacZ}$ brain was significantly decreased compared to WT (Fig. 6d), which was not the case at earlier stages (Supplementary Fig. 4e). CD11b$^+$ cells recovery was also reduced (WT: 140.96 ± 0.91 × 10$^3$, $n = 9$; $Lyl$-$1^{LacZ/LacZ}$: 87.18 ± 0.37 × 10$^3$, $n = 9$). Consequently, $Lyl$-$1$-deficiency triggered a nearly 2-fold reduction of the microglia population (Fig. 6d). This perinatal reduction of microglia appeared transient, since no difference with WT brain was observed in the adult (Supplementary Fig. 4f). Transient decreases of microglia pool size, such as those we observed at E12 and P0-P3 in $Lyl$-$1^{LacZ/LacZ}$ mutant, have been reported to occur during normal development in postnatal weeks 2–3[52], but also in $Cx3cr1$ mutant mice during the 1$^{st}$ postnatal week[53]. This indicates a highly dynamic control of the microglia pool size during key steps of neural development that seems preserved in $Lyl$-$1$ mutant, with the exception of the E12 and P0-P3 time-points. At this later stage, the reduction of brain cellularity in $Lyl$-$1^{LacZ/LacZ}$ mice points to Lyl-1 as a possible regulator of the trophic function of microglia on brain cells[54,55].

The identification of E12 and P0-P3 as key stages for Lyl-1 function in microglia development was confirmed by RT-qPCR analyses of the expression of genes essential for MΦs ($Spi1$/PU.1, $Csf1r$, $Mafb$) and/or microglia ($Runx1$, $Cx3cr1$, $Irf8$) development and function, of known regulators of developmental hematopoiesis ($Tal$-$1$, $Lmo2$, $Runx1$) and related factors ($Tcf3$/E2A, $Tcf4$/E2.2) (Fig. 6e, f; Supplementary Fig. 4g). Time-course analyses highlighted the down-regulation of $Csf1r$, $Irf8$ and $Lmo2$ in $Lyl$-$1^{LacZ/LacZ}$ microglia at E12 and P0-P3, while $Cx3cr1$ was only

decreased at E12 (Fig. 6g). Note that $Lyl$-$1$ expression was unmodified in $Cx3cr1^{GFP/GFP}$ mutants (Fig. 6g). Interestingly, $Cx3cr1$, as well as $Irf8$ and $Lmo2$, belong to potential Lyl-1 target genes[2].

$Mafb$ expression levels in $Lyl$-$1^{LacZ/LacZ}$ microglia transiently decreased at P0-P3 and later returned back to WT expression levels (Fig. 6h). As Mafb represses resident-MΦ self-renewal[56], the recovery of a normal amount of microglia after birth may stem from this transient decrease. $Spi1$/PU.1, $Tcf3$/E2A and $Tcf4$/E2.2 expression levels were unmodified in $Lyl$-$1^{LacZ/LacZ}$ microglia, while $Runx1$ expression was only affected after birth. $Tal$-$1$ expression was decreased at E14 and increased after birth, suggesting that this $Lyl$-$1$ paralog[3] does not compensate $Lyl$-$1$-deficiency during embryonic stages, but may do so at postnatal stages (Supplementary Fig. 4g). Remarkably, RNA-seq. results indicated that some genes deregulated in $Lyl$-$1^{LacZ/LacZ}$ microglia at E12 and P0-P3 were also down-regulated in $Lyl$-$1^{LacZ/LacZ}$ MΦ-progenitors at E9 ($Csf1r$: Supplementary Fig. 3c, $Lmo2$: Fig. 3b; $Irf8$: Fig. 4D; $Cx3cr1$: Fig. 6i). These deregulations were transient, however in both locations and stages they coincided with a defective MΦ/microglia differentiation.

Other genes enriched in microglia ($Fcrls$, $Mef2c$, $Maf$)[57] or involved in the maintenance of microglia homeostasis ($P2ry12$)[58] were also expressed in E9 $Lyl$-$1^{LacZ/LacZ}$ MΦ-progenitors at a lower level than in the WT, except $Lpr8$ and $Aif$-$1$/Iba1 (Fig. 6i). Interestingly, the lower expression of phagocytosis genes ($Axl1$, $Mertk$, $Trem2$, $P2y6r$/$P2ry6$) uncovered in $Lyl$-$1^{LacZ/LacZ}$ MΦ-progenitors from E10-YS (Fig. 4G), was present and stronger in the microglia from $Lyl$-$1^{LacZ/LacZ}$ newborn, except for $Trem2$ which expression was unmodified (Fig. 6j).

These deregulations highlight again shared features between MΦ$^{Prim}$ progenitors and microglia/neural development which became apparent upon $Lyl$-$1$ inactivation considering the large number of neural signaling pathways up-regulated in E9 $Lyl$-$1^{LacZ/LacZ}$ MΦ-progenitors (Fig. 3c) and the relationship of the differentially expressed genes enriched in E9 MΦ$^{Prim}$ progenitors with brain formation and neuro-development uncovered in IPA analysis (Supplementary Fig. 4h).

Based on the gene expression pattern of $Lyl$-$1$-deficient microglia and the signature of MΦ$^{Prim}$ progenitors in the early YS, a contribution of $Lyl$-$1$-deficiency to neurodevelopmental disorders may be considered. Synaptic pruning and neural maturation, which characterize the perinatal phase of microglia development[45], might be impaired in $Lyl$-$1^{LacZ/LacZ}$ embryos

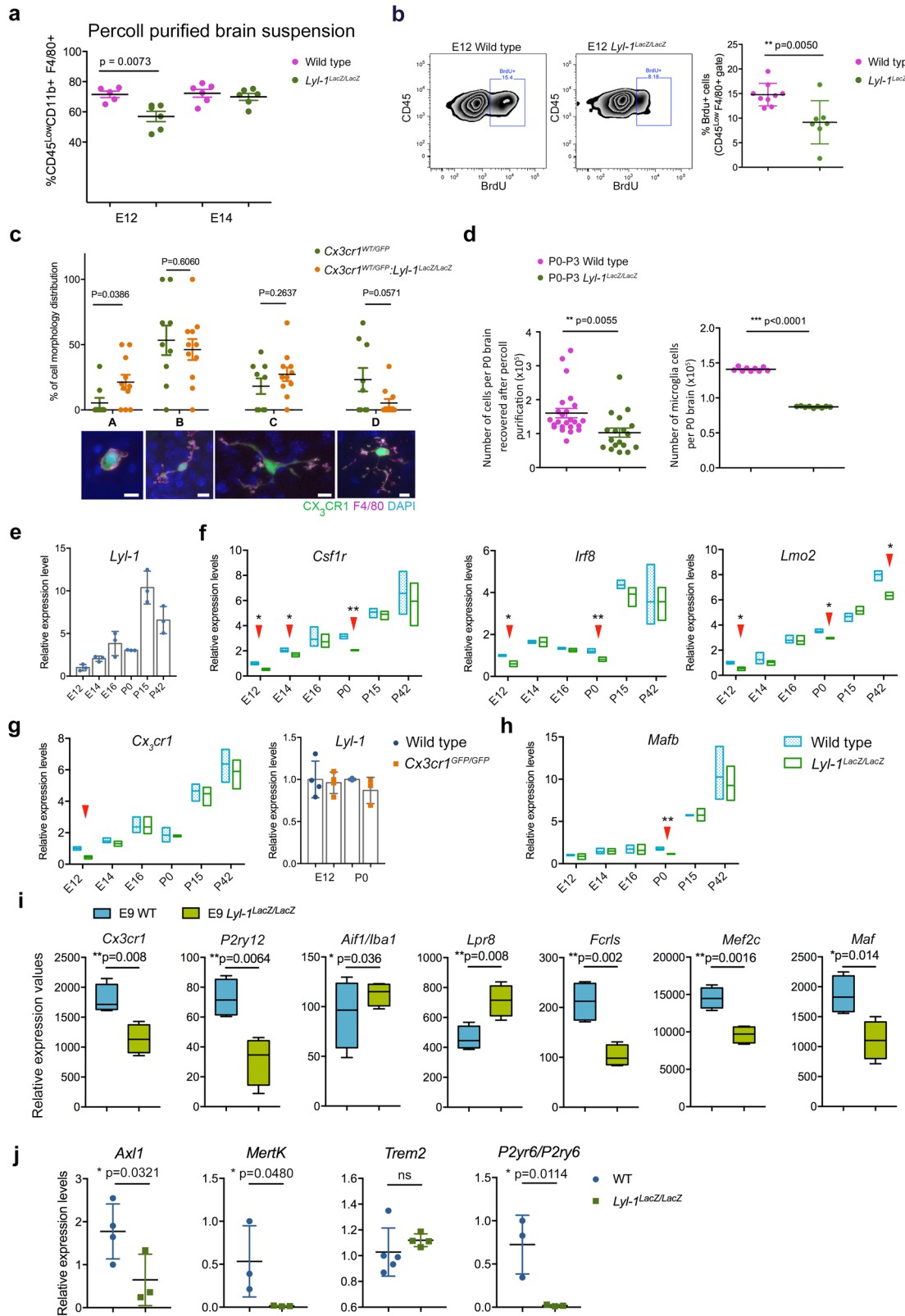

considering the defects observed at P0-P3, the later key developmental stage regulated by Lyl-1. Indeed, *Lyl-1* deregulation has been observed in datasets reporting pathological models of brain myeloid cells (http://research-pub.gene.com/BrainMyeloidLandscape/#Mouse-gene/Mouse-gene/17095/ geneReport.html)[59], as well as in human neuropathies[60,61], including the 19p13.13 micro-deletion neuro-developmental disabilities[62]. However, since Lyl-1 is expressed in endothelial cells, including in the brain[42], a contribution of *LYL-1*-deficient endothelial cells to these diseases must be considered.

**Fig. 6 *Lyl-1* deficiency leads to transient reductions of the microglia pool at E12 and P0-P3. a** Quantification of the microglia population in E12 and E14 brain showing the decreased size of the microglia pool at E12 and its recovery to a normal pool size at E14. (Each dot represents an independent experiment. Error bars show mean ± s.e.m.; two tailed, unpaired *t*-test.) **b** The decreased microglia pool at E12 may result from a reduced proliferation, as shown by the two folds decrease of BrdU-labeled cells in *Lyl-1^{LacZ/LacZ}* compared to WT brains. (4 independent experiments; error bars show mean ± s.e.m.; two tailed, unpaired Mann–Whitney test). **c** At E12, *Cx3cr1^{WT/GFP}:Lyl-1^{LacZ/LacZ}* microglia displayed a reduced number and extent of ramifications compared to their *Cx3cr1^{WT/GFP}* counterpart. Microglia morphology was classified into subtypes depending on the number of main ramifications (A: none, B: 2, C: 3 and D: > 3). Microglia deprived of ramifications predominated in *Lyl-1*-deficient microglia. 65 and 61 cells were respectively acquired from the midbrain of E12 *Cx3cr1^{WT/GFP}* and *Cx3cr1^{WT/GFP}:Lyl-1^{LacZ/LacZ}* embryos (for each genotype, brains from 12 embryos were acquired in 3 independent experiments). Microglia were identified by *Cx3cr1*-driven GFP expression and F4/80-APC immuno-staining. Bar = 10 μm. (Error bars show mean ± s.e.m.; two tailed, unpaired *t*-test). **d** In *Lyl-1^{LacZ/LacZ}* newborns, the cellularity of the brain was consistently lower than in WT, and so was the estimated microglia number. (3 independent experiments. Error bars show mean ± s.e.m.; two tailed, unpaired *t*-test). **e** Kinetic evolution of *Lyl-1* expression levels in WT microglia from embryonic stages to adulthood. Expression levels were normalized to the mean value obtained for E12 microglia (n = 3; error bars show mean ± s.e.m.) An increased expression of *Lyl-1* from embryonic stages to adulthood was also inferred from timeline RNA-seq. data[45]. (GEO accession number GSE79812). **f** Quantitative RT-PCR analyses also point to E12 and P0 as key development stages regulated by Lyl-1. CD11b+F4/80+CD45^{low} microglia were isolated at sequential development stages. Bar graphs show the kinetic of expression of genes modified in *Lyl-1^{LacZ/LacZ}* microglia (arrowheads), normalized to the mean expression value in WT E12 microglia (n = 3). Floating bars indicate mean ± s.e.m. Two tailed, unpaired *t*-test. **g** *Cx3Cr1* and *Lyl-1* expression in mutant microglia. The expression level of *Cx3CR1*, analyzed as in **f**, was decreased in *Lyl-1* mutants at E12, while *Lyl-1* expression levels were unmodified in *CX3CR1^{GFP/GFP}* microglia at E12 and in newborns. **h** *Mafb* expression in mutant microglia. *Mafb* expression level, analyzed as in **f**, was reduced in the microglia of *Lyl-1^{LacZ/LacZ}* newborns. **i** The expression of genes enriched in microglia and/or essential for their function are deregulated in *Lyl-1^{LacZ/LacZ}* MΦ-progenitors at E9. Relative expression levels (read counts) in WT and *Lyl-1^{lacZ/lacZ}* MΦ-progenitors from E9-YS. (3 independent experiments. Box and whisker plots, min to max; median is shown; Unpaired, two-tailed *t*-Test). **j** Relative expression levels (read counts) of phagocytosis genes in sorted microglia from WT and *Lyl-1^{LacZ/LacZ}* newborn (Error bars show mean ± s.e.m.; one tailed unpaired *t*-test).

| Table 1 Breeding crosses used throughout this study. | | |
|---|---|---|
| **Genotype of the male** | **Genotype of the female** | **Genotype of resulting embryos-mice** |
| *Lyl-1^{LacZ/LacZ}* | WT | *Lyl-1^{WT/LacZ}* |
| *Lyl-1^{LacZ/LacZ}* | *Lyl-1^{LacZ/LacZ}* | *Lyl-1^{LacZ/LacZ}* |
| *CX3CR1^{GFP/GFP}* | WT | *CX3CR1^{WT/GFP}* |
| *CX3CR1^{GFP/GFP}* | *Lyl-1^{LacZ/LacZ}* | *Cx3cr1^{WT/GFP}:Lyl-1^{WT/LacZ}* |
| *Lyl-1^{LacZ/LacZ}:CX3CR1^{GFP/GFP}* | WT | *Cx3cr1^{WT/GFP}:Lyl-1^{WT/LacZ}* |
| *Lyl-1^{LacZ/LacZ}:CX3CR1^{GFP/GFP}* | *Lyl-1^{LacZ/LacZ}* | *Lyl-1^{LacZ/LacZ}:CX3CR1^{WT/GFP}* |

| Table 2 Cytokines used in this study. | | |
|---|---|---|
| **Cytokine** | **Concentration** | **Supplier** |
| Murine recombinant stem cell factor | 50 ng/mL | Peprotech; Ref: 315-03 |
| Human recombinant EPO | 3 U/mL | In house production |
| Murine recombinant IL-3 | 10 ng/mL | Peprotech; Ref: 213-13 |
| Human recombinant IL-6 | 10 ng/mL | A gift from Sam Burstein, Maryville, USA |
| Human recombinant CSF-1 | 10 ng/mL | Peprotech; Ref: 300-25 |
| Human recombinant TPO | 10 ng/mL | A gift from Kirin Brewery, Tokyo, Japan |

## Conclusions

Altogether, our findings reveal Lyl-1 as a key factor regulating the production and differentiation of YS MΦ-progenitors and the development of microglia. Lyl-1 is the least studied partners of the transcription factor complex that regulates developmental hematopoiesis. The development of more appropriate models is required to precise Lyl-1 functions in microglia and determine its role in the development of other resident-MΦs populations.

## Methods

**Mice**. The following mouse strains, housed in Gustave Roussy Institute animal facility (License #H94-076-11) were used (See Table 1 for a summary of the breeding schemes): 1- C57BL/6 mice (Harlan or Charles Rivers Laboratories), referred to as wild type (WT); 2- *Lyl-1^{LacZ}* mice[5]. To avoid the possible detection of FDG/Lyl-1 expression from maternally-derived MΦ[11] in heterozygous embryos, *Lyl-1^{LacZ/LacZ}* males were crossed with WT females; 3- *Cx3cr1^{GFP}* mice[63]. *Cx3cr1^{GFP/GFP}* males were crossed with WT females to generate *Cx3cr1^{WT/GFP}* mice/embryos or with *Lyl-1^{LacZ/LacZ}* females to generate *Cx3cr1^{WT/GFP}:Lyl-1^{WT/LacZ}* mice/embryos. 4-*Cx3cr1^{GFP/GFP}:Lyl-1^{LacZ/LacZ}* double mutant strain developed from *Cx3cr1^{WT/GFP}:Lyl-1^{WT/LacZ}* crosses. *Cx3cr1^{WT/GFP}:Lyl-1^{WT/LacZ}* and *Cx3cr1^{WT/GFP}:Lyl-1^{LacZ/LacZ}* mice/embryos were obtained by crossing *Cx3cr1^{GFP/GFP}:Lyl-1^{LacZ/LacZ}* males respectively to C57BL/6 or *Lyl-1^{LacZ/LacZ}* females. Experiments were conducted in compliance with French/European laws, under authorized project #2016-030-5798, approved by officially accredited local institutional animal (committee n°26) and French "Ministère de la Recherche" ethics boards..

The day of vaginal plug observation was considered as E0.5. Pregnant females were sacrificed by cervical dislocation. Pre-somite embryos were staged according to Downs et al.[64], by somite counting from E8 to E10.5 and thereafter by morphological landmarks.

**Tissues**. Cells from the yolk sac (E7.5-E10.5), from E10 FL or whole brain (E9-E14) were obtained after mechanical disruption[9,65]. For cytometry analyses, the dissected region of the brain comprises the di-, mes- and metencephalon at E9-E10, the ectoderm was carefully removed[22]. After E12, microglia were recovered from the midbrain following Percoll (P1644, Sigma) separation[66]. Cells were filtered through a 70 μm cell strainer and centrifuged at 300 g for 10 min. Cells were resuspended in Fc block BD solution and incubated at room temperature for 10 min. After washing in PBS + 10% FCS and centrifugation, cells were resuspended in 50 μl + 10% FCS containing the fluorochrome coupled antibodies and incubated at 4 °C for 20 min. After washing in PBS + 10%FCS and centrifugation, cells were resuspended in 50ul + 10% FCS. Dead cells were excluded by adding 1 μg/mL DAPI (Sigma) before acquisition.

The number of microglia per brain was estimated by reporting the percentage of CD11b+CD45^{low}F4/80+ microglia to the cellularity of the corresponding sample.

**Tissue culture**. YS explants, maintained for 1 day in organ culture in plates containing OptiMEM+Glutamax, 1% Penicillin-streptomycin, 0.1% β-mercaptoethanol (ThermoFisher) and 10% fetal calf serum (Hyclone), are referred to as OrgD1-YS.

In clonogenic assays, YS suspension or sorted cells were plated in triplicate (3 × 10^3 or 100-150 cells/mL) in Methocult® M3234 (StemCell Technologies) always supplemented the cytokines listed in Table 2. Colonies were scored at day 5 for primitive erythroblasts and day 7 for other progenitors.

**Flow cytometry**. Cells, stained with antibodies listed in Table 3, for 30 min. on ice, were acquired (Canto II) or sorted (FACS-Aria III or Influx, BD Biosciences) and analyzed using FlowJo (Treestar) software.

The β-Galactosidase substrate fluorescein di-β-galactopyranoside (FDG; Molecular probe), was used as reporter for Lyl-1 expression in FACS-Gal assay[67,68]. For apoptosis analysis, microglia were immune-stained and incubated with Annexin V-FITC. 7AAD was added before acquisition. For proliferation assays, pregnant females (12 gestational days) were injected with BrdU (10μM) and

**Table 3 Antibodies and fluorescent stains used throughout this study.**

| Antibody name | Clone | Supplier/Reference | Fluorochrome/ Chromogen |
|---|---|---|---|
| Ter119 | TER-119 | Biolegend; 116205 | FITC |
| | | Biolegend; 116208 | PE |
| | | eBioscience 17-5921-82 | APC |
| F4/80 | BM8 | eBioscience; 53-4801-82 | Alexa fluor 488 |
| | CI:A3-1 | Biolegend; 122606 | FITC |
| | BM8 | eBioscience; 12-4801-82 | PE |
| | BM8 | Biolegend; 123116 | APC |
| GR-1 | RB6-8C5 | BD-Pharmingen; 553127 | FITC |
| | RB6-8C5 | Biolegend; 108416 | PE-Cy7 |
| | RB6-8C5 | Biolegend; 108437 | BV510 |
| CD45 | 30-F11 | BD-Pharmingen; 553081 | PE |
| | 30-F11 | BD-Pharmingen; 553082 | PE-Cy5 |
| | 30-F11 | eBioscience; 25-0451-82 | PE-Cy7 |
| | 30-F11 | Biolegend; 103124 | Alexa fluor 647 |
| CD31 | MEC13.3 | Biolegend; 102514 | Alexa fluor 488 |
| | 390 | Biolegend; 102408 | PE |
| | MEC13.3 | Biolegend; 102419 | PE-Cy5.5 |
| | MEC13.3 | Biolegend; 102516 | Alexa fluor 647 |
| | MEC13.3 | BD-Pharmingen; 583089 | BV510 |
| Kit | 2B8 | Biolegend; 05824 | Alexa fluor 488 |
| | 2B8 | BD-Pharmingen; 553358 | APC |
| | 2B8 | Biolegend; 105825 | APC-Cy7 |
| CD11b | M1/70 | eBioscience; 45-0112-82 | PE-Cy5.5 |
| | | Biolegend; 101215 | PE-Cy7 |
| | | Biolegend; 101212 | APC |
| | | eBioscience; 47-0112-82 | APC-eFluor 780 |
| Sca-1 | D7 | Biolegend; 108113 | PE-Cy7 |
| MHC-II | M5/114.15.2 | Biolegend; 107607 | PE |
| Annexin V | NA | Biolegend; B206040 | FITC |
| Anti-Brdu | B44 | BD-Pharmingen; 51-23619 L | APC |
| 7AAD | NA | BD-Pharmingen; 51-68981E | Dye |
| FDG | NA | Molecular probe; Thermo Fisher; F1179 | Dye |

sacrificed 2 h later. Microglia were isolated, immune-stained and prepared according to kit instruction (BD Pharmingen 552598). BrdU incorporation was revealed using anti-BrdU-APC.

**Brain imaging**. To assess microglia morphology in E12 embryos, the midbrain was dissected from $Cx3cr1^{WT/GFP}:Lyl-1^{WT/WT}$ and $Cx3cr1^{WT/GFP}:Lyl-1^{LacZ/LacZ}$ embryos and sectioned through the midline. After fixation (4% paraformaldehyde) overnight at 4 °C, whole midbrains were washed in phosphate-buffered saline (PBS)/0.1 M glycine and incubated overnight in PBS/15% sucrose at 4 °C. Midbrains were washed with PBS + 0.1% Tween and incubated 90 min. in blocking buffer (PBS + 10% FCS) at room temperature (RT). Midbrains were subsequently immune-labeled with F4/80-APC overnight at 4 °C. After washing, they were incubated 3 min. in PBS + DAPI (1 µg/mL) at RT and washed. Finally, midbrains were placed in the central well of glass-bottom culture dishes (P35G-1.5-10-C; MatTek, USA) filled with PBS + 10% FCS. After appropriate orientation of the sample, the well was covered with a 12 µm ∅ glass coverslip. Image stacks were collected using a Leica SP8 confocal microscope. To unsure an unbiased choice of the cells imaged, taking into account possible changes in cell distribution induced by *Lyl-1* deficiency, we always acquired cells in similar positions regarding the landmark set in the midbrain flat mount, as shown in Supplementary data and Fig. 4C. Images were processed using Imaris x64 (version 7.7.2; Bitplane) and Photoshop 8.0 (Adobe Systems, San Jose, CA) softwares.

**RT-qPCR analyses**. RNA was extracted using Trizol. After cDNA synthesis (SuperScript™ VILO™ Master-Mix reverse transcriptase, ThermoFisher), Real Time (RT)-PCR was performed (SYBR Premix Ex TaqII, Takara Bio). Reference genes were *Actin*, *Hprt* and *Tubulin*. Gene expressions were normalized to the values obtained for E10-YS MΦ-progenitors, E12 WT Lin⁻Sca⁺cKit⁺ (LSK) progenitors or E12 WT microglia. Primers are listed in Table 4.

**RNA-seq**. For samples preparation, MΦ-progenitors (Kit⁺CD45⁺CD11b⁺) were sorted from E9 (MΦ^Prim progenitor) and E10 (MΦ^Prim + MΦ^T-Def progenitors) YS pools from WT or *Lyl-1*^LacZ/LacZ embryos (Four biological replicates). RNA was extracted as described above.

The RNA integrity (RNA Integrity Score ≥ 7.0) was checked on the Agilent 2100 Bioanalyzer (Agilent) and quantity was determined using Qubit (Invitrogen).

SureSelect Automated Strand Specific RNA Library Preparation Kit was used according to manufacturer's instructions with the Bravo Platform. Briefly, 50 ng of total RNA sample was used for poly-A mRNA selection using oligo(dT) beads and subjected to thermal mRNA fragmentation. Fragmented mRNA samples were subjected to cDNA synthesis and converted into double stranded DNA using reagents supplied in the kit. The resulting dsDNA was used for library preparation. The final libraries were bar-coded, purified, pooled together in equal concentrations and subjected to paired-end sequencing (2 × 100) on Novaseq-6000 sequencer (Illumina) at Gustave Roussy genomic facility.

The quality of RNA-seq. reads were assessed with FastQC 0.11.7 and MultiQC 1.5[69]. Low quality reads were trimmed with Trimmomatic 0.33[70]. Salmon 0.9.0 tool[71] was used for quantifying the expression of transcripts using geneset annotation from Gencode project release M17 for mouse[72]. The version of transcriptome reference sequences used was GRCm38.p6.

Statistical analysis was performed using R with the method proposed by Anders and Huber implemented in the DESeq2 Bioconductor package[73]. The differential expression analysis in DESeq2 uses a generalized linear model (GLM) where counts are modeled using a negative binomial distribution. Counts were normalized from the estimated size factors using the median ratio method and a Wald test was used to test the significance of GLM coefficients. Genes were considered differentially expressed when adjusted *p*-value < 0.05 and fold-change > 2.

Data were analyzed through the use of Ingenuity® Pathway Analysis (QIAGEN Inc., https://www.qiagenbioinformatics.com/products/ingenuity-pathway-analysis)[74], Gene set enrichment analysis (GSEA; https://www.gsea-msigdb.org/gsea/index.jsp)[75,76], Morpheus (https://software.broadinstitute.org/morpheus/) and Venny (https://bioinfogp.cnb.csic.es/tools/venny/) softwares.

**Statistics and reproducibility**. Statistical analyses including mean, SEM and *p* values were performed using using Prism 7 software (GraphPad). Statistical significance between two groups was assessed by unpaired two-tailed Student's *t* tests or Mann–Whitney tests. Data are presented as mean ± SEM or by Box and whisker plots, min to max, with the median shown. Statistical significances were indicated by *p*-values and/or as *$p < 0.05$, **$p < 0.01$, ***$p < 0.001$, and ****$p < 0.0001$, ns: not significant. *P*-values below 0.05 were considered as significantly different. The number of biological replicates (*n*) and independent experiments is reported in the corresponding figure legend.

**Table 4 Primers used in this study.**

| Gene | | Sequence |
|---|---|---|
| Actin | Primer forward: | 5′- CTTCTTTGCAGCTCCTTCGT-3′ |
|  | Primer reverse: | 5′- ATCACACCCTGGTGCCTAG-3′ |
| Hprt | Primer forward: | 5′- TGATTATGGACAGGACTGAAAGA-3′ |
|  | Primer reverse: | 5′- AGCAGGTCAGCAAAGAACTTATAG-3′ |
|  | Primer reverse: | 5′-TGGTTCTGCATCGACTTCTG-3′ |
| Tubulin | Primer forward: | 5′- GGGTGGAGGCACTGGCT-3′ |
|  | Primer reverse: | 5′- GTCAGGATATTCTTCCCGGATCT-3′ |
| c-Maf | Primer forward: | 5′- CTGCCGCTTCAAGAGGGTGCAGC -3′ |
|  | Primer reverse: | 5′- GATCTCCTGCTTGAGGTGGTC -3′ |
| Lyl-1 | Primer forward: | 5′- TGGACTGACAAACCTGACCA-3′ |
|  | Primer reverse: | 5′- TGGACCCCACGGATAGAATG-3′ |
| Tal-1/Scl | Primer forward: | 5′- CATGTTCACCAACAACAACCG-3′ |
|  | Primer reverse: | 5′- GGTGTGAGGACCATCAGAAATCTC-3′ |
| Runx1 | Primer forward: | 5′- CTCCGTGCTACCCACTCACT-3′ |
|  | Primer reverse: | 5′- ATGACGGTGACCAGAGTGC-3′ |
| Meis1 | Primer forward: | 5′-TGAAGTAGGAAGGGAGCCAG-3′ |
|  | Primer reverse: | 5′-GCCTACTCCATCCATACCCC-3′ |
| Lmo2 | Primer forward: | 5′-ATGTCCTCGGCCATCGAAAG-3′ |
|  | Primer reverse: | 5′-CGGTCCCCTATGTTCTGCTG-3′ |
| PU.1 | Primer forward: | 5′- GCTATACCAACGTCCAATGCA-3′ |
|  | Primer reverse: | 5′-TGTGCGGAGAAATCCCAGTA-3′ |
| Tcf3/E2A | Primer forward: | 5′-GGGCTCTGACAAGGAACTGA-3′ |
|  | Primer reverse: | 5′-AGTCCTGAGCCTGCAAACTG-3′ |
| Tcf4/E2.2 | Primer forward: | 5′-GCCTCTTCACAGTAGTGCCAT-3′ |
|  | Primer reverse: | 5′-TCCCTGTTGTAGTCGGCAGT-3′ |
| Csf1r | Primer forward: | 5′-GCAGTACCACCATCCACTTGTA-3' |
|  | Primer reverse: | 5′-GTGAGACACTGTCCTTCAGTGC-3' |
| Cx3cr1 | Primer forward: | 5′ - AGTTCCCTTCCCATCTGCTC-3′ |
|  | Primer reverse: | 5′ - GCCACAATGTCGCCCAAATA-3′ |
| Irf8 | Primer forward: | 5′- CGGGGCTGATCTGGGAAAAT-3′ |
|  | Primer reverse: | 5′- CACAGCGTAACCTCGTCTTC-3′ |
| Myb | Primer forward: | 5′-AGCGTCACTTGGGGAAAACT-3′ |
|  | Primer reverse: | 5′- AGTCGTCTGTTCCGTTCTGT-3′ |
| Axl | Primer forward: | 5′-TGAGCCAACCGTGGAAAGAG-3′ |
|  | Primer reverse: | 5′-AGGCCACCTTATGCCGATCTA-3′ |
| MertK | Primer forward: | 5′-GATTCTGGCCAGCACAACAGA-3′ |
|  | Primer reverse: | 5′-GAGATATCCGGTAGCCCACCA-3′ |
| Trem2 | Primer forward: | 5′- CTGGAACCGTCACCATCACTC-3′ |
|  | Primer reverse: | 5′- CGAAACTCGATGACTCCTCGG-3′ |
| P2y6r/P2ry6 | Primer forward: | 5′-GCGTCTACCGTGAGGATTTCA-3′ |
|  | Primer reverse: | 5′-CCACCAGCACCACCGAGTAT-3′ |

**Reporting summary**. Further information on research design is available in the Nature Research Reporting Summary linked to this article.

## Data availability

The authors declare that all data supporting the findings of this study are available within the article and its supplementary information files or from the corresponding author upon reasonable request. RNAseq data have been deposited in the ArrayExpress database at EMBL-EBI (www.ebi.ac.uk/arrayexpress) under accession code E-MTAB-9618. The source data underlying figures and supplementary figures are provided as a Supplementary Data 1.

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

## Acknowledgements

The authors thank Julien Bertrand for critical reading of the manuscript. We are grateful to the staff of the facilities at Gustave Roussy, the animal facility (PFEP, UMS AMMICa UMS 3655/US23, directed by P. Gonin), the imaging facility (PFIC, UMS AMMICa UMS 3655/US23, directed by C. Laplace-Builhe), the genomic facility directed by N. Droin, the bioinformatics facility (G. Meurice), directed by M. Deloger. This work was supported by fundings from Institut National de la Santé et de la Recherche Médicale to W. Vainchenker, I. Plo and H. Raslova, from Centre National de la Recherche Scientifique and Université de Paris-Saclay to I. Godin, from grants INCA PLBio to I. Plo, "Ligue Nationale contre le Cancer" Certified Team to H. Raslova, "Association pour la Recherche sur le Cancer" (n°4878) to I. Godin, Gustave Roussy (TA DERE 17) and National Natural Science foundation of China (N°32000669) to D. Ren, Grant Agency of the Czech Republic (GACR n°19–23154 S) to D. Filipp and from fellowships from "Association pour la Recherche sur le Cancer" to A.-L. Kaushik; "Société Française d'Hématologie" to S. Wang and Chinese Scholarship Council fellowships to S. Wang and D. Ren.

## Author contributions

SW: Conceptualization; Methodology; Validation; Formal analysis; Investigation; Data curation; Writing – original draft preparation; Visualization; DR: Formal: analysis; Investigation; Data curation; Writing – original draft preparation; Visualization; A-LK; BA: Investigation, methodology; Formal analysis; Visualization; GM: Formal analysis; Investigation; methodology; YL: Investigation; methodology; DF: Writing—review & editing: critical review, commentary and revision; WV, HR, IP: Funding acquisition; Writing—review & editing: critical review, commentary and revision; IG: Conceptualization; Project administration; Supervision; Methodology; Validation; Formal analysis; Resources; Data curation; Writing – original draft preparation; Writing—review & editing; Visualization; Funding acquisition.

## Competing interests

The authors declare no competing interests.
