## [Peer Review File · Communications Biology]

Reviewers' comments:

Reviewer #1 (Remarks to the Author):

Wang et. al. described Lyl-1 is expressed in yolk-sac derived macrophage(MF) progenitors (M ϕ prim progenitor), which have distinct transcriptomic features of other MF progenitors (M ϕ T-Def progenitors). This study is important in the field of macrophage development, as any genetic models distinguish early-EMP from late-EMP yet. Although authors show Lyl-1 deficiency showed the defects of embryonic patterning, of M ϕ prim progenitor and microglia development using Lyl-1 knock-out (Lyl-lacZ/lacZ) mice and their controls, I do have a few concerns especially in RNA-seq analysis as addressed below.

Major concerns,

- Authors showed macrophage progenitor colonies (%) in organ cultures obtained from yolk sac of WT, Lyl-1 heterozygotes, Lyl-1 knockout mice. It was not shown what markers were used to define the macrophage colonies, GMP, EMP. Accordingly, authors need to show these population (e.g. flow cytometric analysis) in parallel with Fig. 1A.
- Authors nicely showed the distinct frequency of macrophage progenitors through the organ culture. As culture system is an artificial setting, we can not exclude the YS-tissue specific factors can recover these phenotypes. Accordingly, authors should show the quantification of M ϕ prim progenitor from WT, Lyl-lacZ/+ and Lyl-lacZ/lacZ mice at E8 stage.
- Authors clearly showed single population of FDG+ M ϕ progenitors at E9 and two distinct M ϕ progenitors (FDG+ and FDG-) at E10 in Fig. 1B and Fig. 1D-E. It raises the questions whether these three progenitors have similar/distinct gene expression patterns (i.e. E9 M ϕ prim progenitor v.s. E10 M ϕ prim (FDG+) progenitor; E10 M ϕ prim progenitor (FDG+) v.s. E10 M ϕ T-Def progenitor (FDG-)) through heatmaps or volcano plot. Accordingly, authors should compare the gene expression among three populations, instead of gene expression of macrophage progenitors between E9 and E10.
- This manuscript addressed the role of Lyl-1 during the macrophage development (Fig. 4-6). However, authors did not compare the gene expression patterns between wild type and Lyl-1lacZ/lacZ mice but showed the gene expression in development stage (E9 and E10) although RNA-seq was performed with these two strains (WT and Lyl-1lacZ/lacZ mice) at E9 and E10 (Fig. 2A). It would be great if authors show the gene expression patterns at E9 and E10 between wild type and Lyl-1lacZ/lacZ mice.
- Authors suggested embryonic patterning defect can be the defective M ϕ prim progenitor (line 213). However, it seems too ambitious interpretation. We are not able to exclude the other possibilities from whole Lyl-1 knock-out mice, as genetic models used in this study not a macrophage-specific conditional Lyl-1 ko out. Authors address it in discussion part.
- Csf1-Csf1R signaling is important in the development of macrophages (i.e. survival, differentiation of macrophages). In Fig. 4C, authors showed Csf1r expression was reduced in E9 Lyl-1 ko mice, implying the down regulation of these genes delays the differentiation of E9 M ϕ prim progenitor. To confirm it whether these genes affect the differentiation progenitors or not, it would be great to compare the gene expression of Csf1r in CD11b+ F4/80+ mature macrophages (E10 or adult) in WT mice and Lyl-1 ko mice.
- Results and discussion was combined in this manuscript. Authors need to split the result part and discussion, as it is not the brief report but an article.

Minor comments.

- In Fig. 1A, graph showed M ϕ progenitors in y-axis but it is written as M ϕ colonies. Authors need to clarify whether it is progenitors or macrophages.
- Statistics were not clearly addressed in methods. How do you generate the statistics in Fig. 4A? Is it ANOVA multiple comparison study? Authors provide the statistics (e.g. ANOVA multiple comparison or student's t test) in each figure legends.
- In addition to review papers (reference 15), it would be better to add the original research paper in line 80.
- It would be great to provide an audience the table for the breeding strategy shown in supplementary method.

Reviewer #2 (Remarks to the Author):

The hematopoietic origin of tissue resident macrophages and, in particular, microglia has been the subject of intense investigation in recent years. Yet, the impact of early hematopoietic heterogeneity in adult macrophage function is less understood. In the present manuscript, Wang et al describe the implication of the transcription factor Lyl-1 in primitive macrophage progenitors and microglia development. This work shows the Lyl-1 dependent gene regulation in microglia and macrophage progenitors at two stages of the embryonic and postnatal development. Particularly interesting is the concept of transient gene regulation dependent on Lyl-1 during macrophage/microglia ontogeny. The submitted article comprises a very detailed study of gene expression changes in progenitors and microglial cells, supported with analysis of the diverse macrophage precursors and microglia at the yolk sac, fetal liver and brain at several developmental stages. Overall, the present manuscript by Wang et al uncovers a novel mechanism influencing microglia development at embryonic and early postnatal stages depending on Lyl-1, potentially applicable to other tissue resident macrophage populations. It has an important potential impact to understand microglia functions during development and its impact in central nervous system organogenesis. Potential improvement of the manuscript may be made by correcting a few typos and rephrasing some paragraphs in the current form of manuscript, in addition to addressing the following points:

1. Although the authors acknowledge this limitation and apply diverse in silico methods to analyse these data, the gene expression signature of E10 populations from the RNA-Seq experiments (Fig 2) comes from a mixed population. Performing sc-RNA-Seq of E9 and E10 CD45+ cells would be extremely useful and might overcome this limitation. Alternatively, spatial analysis of a collection of target genes/proteins identified in the current RNA-Seq analysis using embryo tissue slices as source material would be really informative.
2. In figure 3, most of the conclusions of how changes in gene expression due to Lyl-1 affect the development of the central nervous system are based on predicted signaling pathways and process derived from the sequencing data. The specific validation of some of these pathways would be desirable. For example, immunohistochemistry or in situ hybridization to detect specific genes regulated by Lyl-1, or histological evaluation of Lyl-1-deficient embryos at E8, E9, E10 to assess pattern organization and the distribution of macrophage progenitors.
3. In figure 6 (and suppl 4) the authors show a decreased rate of apoptosis and proliferation of Lyl-1-deficient microglia at E12, which might very well explain the similar frequency of microglial cells at E14, however, at E12, the population of Lyl-1-deficient microglia is nevertheless reduced (Fig 6a). Interestingly, Lyl-1-deficient microglia shows dramatic morphological changes at day E12, with deprivation of ramifications. Although the non-ramified microglia seem to preserve intact membrane (as of F4/80 staining) the authors should test whether those are late apoptotic cells non detectable by FACS measurement of AnnexinV (for example using TUNEL reaction and F4/80 staining).
4. The presence of these aberrant microglial cells is per se really interesting, and their functional implications should be further analyzed. Do these non-ramified microglial cells show defects in phagocytosis or neurons or environmental sampling, for example? Possibly in vitro approaches using neonatal-derived microglia could address this point.

Reviewer #3 (Remarks to the Author):

Production of selected immune cells, including macrophages, relies before the generation of HSC on two consecutive hematopoietic waves, primitive and transient definitive, that both arise in the YS and have also been termed 'early' (eEMP) and 'late' EMP. Most tissue macrophages, including microglia, are now appreciated to originate from these early waves, and subsequently self-maintain independent from HSC input. Differential contributions of eEMP and EMP to the macrophage compartments remain however less well defined. Here the authors focus on the BLH transcription factor (TF) Lyl-1 (1) as marker for early primitive macrophage precursors in the YS, and (2) with respect to its function in microglia development.

In the first part the authors take advantage of published Lyl-1-lacZ reporter animals, in combination with the FDG assay, to establish that Lyl-1 expression allows to discriminate primitive from transient-definitive macrophage precursors in the YS. They then move on to compare differentiation potential of E9 and E10 MF progenitors, including short term YS organ cultures, flow cytometry, RNAseq analysis and clonogenic assays. The accumulation of MF precursors in the E8 Lyl-1 deficient YS suggests that in WT mice the TF restricts the progenitor pool size potentially by affecting the window of production.

The authors further report that the Lyle-1 deficiency results in a transient decrease of the microglia pool size. Since microglia themselves also express Lyle-1, it remains unclear whether this phenotype is due to affected primitive progenitors or direct effects on microglia.

Taken together, this is an interesting study addressing an important biological question. The conclusions were found largely supported by solid data, although the study is largely descriptive and statements are hence often based on correlations rather than formal proof of causality. The manuscript could be significantly improved in clarity and a clearer formulation of its key findings. At the moment the reader is left with a number of interesting observations, but the main message remains somewhat elusive because elaborations are limited.

Specific points

The authors seem to claim to provide evidence that microglia arise from early EMP. They base this conclusion on transcriptome overlap, including expression of the microglia signature gene Sall1. However, this part of the study, which would represent a major finding, came across as underdeveloped and could be carved out better.

The authors mention MHC II expression at E9 and E10 (line 169), addition of a representative FACS blot would here be convincing.

The authors note limited differentiation of E9 MF, and mention evidence in the sequencing data but this could be elaborated further. In line 238, they

The authors mention defective cytokine signaling (line 238), but it remains unclear how they got to that conclusion, Suppl. Fig 3D/E is here rather unclear.

The authors mention the identification of a Lyl-1 core signature, but don't elaborate further.

The contribution of Lyl-1 to the BAM compartment could be expanded, by simply adding a FACS blot showing CD11b/ F4/80 or CD206/Cx3Cr1 (see Utz et al, ref 16). Are BAM affected in the Lyl-1 KO?

Answer to the reviewers

Reviewer #1: Wang et. al. described Lyl-1 is expressed in yolk-sac derived macrophage(MF) progenitors (M ϕ prim progenitor), which have distinct transcriptomic features of other MF progenitors (M ϕ T-Def progenitors). This study is important in the field of macrophage development, as any genetic models distinguish early-EMP from late-EMP yet. Although authors show Lyl-1 deficiency showed the defects of embryonic patterning, of M ϕ prim progenitor and microglia development using Lyl-1 knock-out (Lyl-lacZ/lacZ) mice and their controls, I do have a few concerns especially in RNA-seq analysis as addressed below.

Major concerns:

- Authors showed macrophage progenitor colonies (%) in organ cultures obtained from yolk sac of WT, Lyl-1 heterozygotes, Lyl-1 knockout mice. It was not shown what markers were used to define the macrophage colonies, GMP, EMP. Accordingly, authors need to show these population (e.g. flow cytometric analysis) in parallel with Fig. 1A.

Answer: Fig. 1A reported the quantification of progenitors functionally identified in clonogenic assay (= CFC, Clonogenic Forming Cell assay). In this assay, the type of progenitors/forming cell is defined by the nature of mature cells contained in the colony derived from each progenitor, spatially separated within the semi-solid medium (methyl cellulose). The cell type of the progeny of each progenitor is characterized using well established morphological criteria. This biological assay is more precise than cytometry analysis particularly at early development stages, when progenitors for the various lineages are rare.

- Authors nicely showed the distinct frequency of macrophage progenitors through the organ culture. As culture system is an artificial setting, we cannot exclude the YS-tissue specific factors can recover these phenotypes. Accordingly, authors should show the quantification of M ϕ prim progenitor from WT, Lyl-lacZ/+ and Lyl-lacZ/lacZ mice at E8 stage.

Answer: We had shown the quantification of E8 M ϕ -progenitors through clonogenic assay in Figure 3A, as an introduction to the RNA-seq. analysis. Thanks to the reviewer remark, we realized that this dataset should more logically appear in Figure 1. It now appears as Figure 1C.

- Authors clearly showed single population of FDG+ Mφ progenitors at E9 and two distinct Mφ progenitors (FDG+ and FDG-) at E10 in Fig. 1B and Fig. 1D-E. It raises the questions whether these three progenitors have similar/distinct gene expression patterns (i.e. E9 Mφprim progenitor v.s. E10 Mφprim (FDG+) progenitor; E10 Mφprim progenitor (FDG+) v.s. E10 MφT-Def progenitor (FDG-)) through heatmaps or volcano plot. Accordingly, authors should compare the gene expression among three populations, instead of gene expression of macrophage progenitors between E9 and E10.

Answer: Following the same lead as reviewer #1, we had included in our RNA-seq analysis FDG positive and negative Mφ progenitor samples, sorted from E10 *Lyl-1^{LacZ/LacZ}* YS. Unfortunately, this part of our study did not provide enlightening information.

Briefly, The PCA plot (A) for all the analyzed samples clearly indicated that the stage is the most stringent discriminating criteria. Focusing on PC1, it appeared that FDG⁺ and FDG⁻ populations, obtained from mutant YS, cluster together with E10 mutant samples and did not separate well from each other. We expected that the FDG⁻ subset would cluster closer to the E10 WT subset and it instead appeared closer to the E10 *Lyl-1^{LacZ/LacZ}* subset Mφ-progenitor samples.

Few genes (23 DEG: B) were identified, 10 enriched in the FDG⁺ subset compared to the FDG⁻ one and 13 enriched in the FDG⁻ subset compared to the FDG⁺ one. Except for a tendency of the DEG to belong to the defective pathways uncovered in E9 vs E10 WT or in E9 WT vs *Lyl-1^{LacZ/LacZ}* subsets, the data obtained were not informative and were therefore not mentioned in the article.

These observations reinforce the interest to construct a new *Lyl-1* reporter strain that would allow a clear-cut separation between *Lyl-1*-positive and negative Mφ progenitors subsets. This strain

would permit to investigate the relative contribution and function of Lyl-1-positive and Lyl-1-negative MΦ populations to the various tissues that harbor resident-MΦ of YS-origin and would be a valuable tool to gain more insight on the relative function of these two MΦ, in the framework of projects aim to assess the importance of pathways differentially expressed in WT and *Lyl-1^{lacZ/lacZ}* MΦ progenitors.

- This manuscript addressed the role of Lyl-1 during the macrophage development (Fig. 4-6). However, authors did not compare the gene expression patterns between wild type and Lyl-1lacZ/lacZ mice but showed the gene expression in development stage (E9 and E10) although RNA-seq was performed with these two strains (WT and Lyl-1lacZ/lacZ mice) at E9 and E10 (Fig. 2A). It would be great if authors show the gene expression patterns at E9 and E10 between wild type and Lyl-1lacZ/lacZ mice.

Answer: Due to the loss of the *Lyl-1^{lacZ/LacZ}* mouse strain, expression pattern analyses could not be performed. We believe that analyzing the expression pattern of few candidates differentially expressed genes in E9 and E10 WT and *Lyl-1^{lacZ/LacZ}* MΦ-progenitors might rise more questions than provide answers. The importance of differentially expressed pathways would require the development of well-designed research projects.

Particularly interesting to investigate would be the inflammatory pathway, prevalent in the primitive population and dependent on Lyl-1 expression, as the function of macrophages in the generation of Hematopoietic Stem cells, together with the importance of inflammatory signaling in this process is now well established.

- Authors suggested embryonic patterning defect can be the defective MΦprim progenitor (line 213). However, it seems too ambitious interpretation. We are not able to exclude the other possibilities from whole Lyl-1 knock-out mice, as genetic models used in this study not a macrophage-specific conditional Lyl-1 ko out. Authors address it in discussion part.

Answer: We improved the discussion by stating that the patterning defects uncovered by the RNA-seq. analysis might result from a defective function of macrophages, but that a contribution of impaired development of other Lyl-1-expressing cell types, such as endothelial cells, should also be considered, leading to the conclusion that only the development of lineage specific Lyl-1 mice models would allow to better understand Lyl-1 function during ontogeny.

-*Csf1-Csf1R* signaling is important in the development of macrophages (i.e. survival, differentiation of macrophages). In Fig. 4C, authors showed *Csf1r* expression was reduced in E9 *Lyl-1* ko mice, implying the down regulation of these genes delays the differentiation of E9 $M\phi^{prim}$ progenitor. To confirm it whether these genes affect the differentiation progenitors or not, it would be great to compare the gene expression of *Csf1r* in $CD11b^+ F4/80^+$ mature macrophages (E10 or adult) in WT mice and *Lyl-1* ko mice.

Answer: The comparison of *Csf1r* expression levels in $F4/80^+$ $M\phi$ purified from E10-11 WT (n=2) and *Lyl-1*^{LacZ/LacZ} (n=4) whole embryos, deprived of Head and YS, did not show a significant decrease of *Csf1r* expression in *Lyl-1* mutant. This lack of significant change might be due to the distribution of mature $M\phi$ in embryonic tissues. While primitive and transient-

definitive $M\phi$ -progenitors co-exist in the embryo E10-11, the location of their mature progeny appears tissue specific, $M\phi^{prim}$ progenitors contributing mainly to brain $M\phi$ (absent from our RT-qPCR samples). Actually, the data illustrating the down-regulation of *Csf1r* in *Lyl-1*^{LacZ/LacZ} microglia at E12, E14 and in the new-born were shown in figure 6F.

-Results and discussion was combined in this manuscript. Authors need to split the result part and discussion, as it is not the brief report but an article.

Answer: We first had written a separate discussion and found it difficult to keep short, simple and meaningful to readers, as this study covers various development stages, tissues and techniques, and as many unsolved issues/risen questions needed to be addressed. We found the paper was easier to read and the conclusions clearer when each point to be discussed was addressed when first risen. As the editor mentioned in the review cover: "If you wish, you can have a separate discussion section", we considered that we could keep it separated.

Minor comments.

- In Fig. 1A, graph showed $M\phi$ progenitors in y-axis but it is written as $M\phi$ colonies. Authors need to clarify whether it is progenitors or macrophages.

Answer: In clonogenic assays, the type of progenitor is determined by the type of hematopoietic cells found in the colony it produces, so that the type of colony and that of the progenitor is equivalent. To simplify this point, we consistently used the word progenitor throughout the paper.

- *Statistics were not clearly addressed in methods. How do you generate the statistics in Fig. 4A? Is it ANOVA multiple comparison study? Authors provide the statistics (e.g. ANOVA multiple comparison or student's t test) in each figure legends.*

Answer: In the revised version, we complemented the description of statistic comparison processes in Material and methods and in each figure legend in the main article and supplemental data.

- *In addition to review papers (reference 15), it would be better to add the original research paper in line 80.*

Answer: The reference for the first mention of “early and late EMPs” qualification for the progenitors of the primitive and transient-definitive wave naming (*Hoeffel et al Immunity. 2015*) was added together with the review (*Ginhoux et al. Immunity 2016*).

- *It would be great to provide an audience the table for the breeding strategy shown in supplementary method.*

Answer: A table reporting the breeding strategy used was added as *Supplementary table 5*.

Reviewer #2: The hematopoietic origin of tissue resident macrophages and, in particular, microglia has been the subject of intense investigation in recent years. Yet, the impact of early hematopoietic heterogeneity in adult macrophage function is less understood. In the present manuscript, Wang et al describe the implication of the transcription factor Lyl-1 in primitive macrophage progenitors and microglia development. This work shows the Lyl-1 dependent gene regulation in microglia and macrophage progenitors at two stages of the embryonic and postnatal development. Particularly interesting is the concept of transient gene regulation dependent on Lyl-1 during macrophage/microglia ontogeny. The submitted article comprises a very detailed study of gene expression changes in progenitors and microglial cells, supported with analysis of the diverse macrophage precursors and microglia at the yolk sac, fetal liver and brain at several developmental stages. Overall, the present manuscript by Wang et al uncovers a novel mechanism influencing microglia development at embryonic and early postnatal stages depending on Lyl-1, potentially applicable to other tissue resident macrophage populations. It has an important potential impact to understand microglia functions during development and its impact in central nervous system organogenesis. Potential improvement of the manuscript may be made by correcting a few typos and rephrasing some paragraphs in the current form of manuscript, in addition to addressing the following points:

1. Although the authors acknowledge this limitation and apply diverse in silico methods to analyse these data, the gene expression signature of E10 populations from the RNA-Seq experiments (Fig 2) comes from a mixed population. Performing sc-RNA-Seq of E9 and E10 CD45+ cells would be extremely useful and might overcome this limitation. Alternatively, spatial analysis of a collection of target genes/proteins identified in the current RNA-Seq analysis using embryo tissue slices as source material would be really informative.

2. In figure 3, most of the conclusions of how changes in gene expression due to Lyl-1 affect the development of the central nervous system are based on predicted signaling pathways and process derived from the sequencing data. The specific validation of some of these pathways would be desirable. For example, immunohistochemistry or in situ hybridization to detect specific genes regulated by Lyl-1, or histological evaluation of Lyl-1-deficient embryos at E8, E9, E10 to assess pattern organization and the distribution of macrophage progenitors.

Answer: Due to the loss of the *Lyl-1^{LacZ/LacZ}* mouse strain, expression pattern analyses could not be performed. We believe that analyzing the expression pattern of few candidates differentially

expressed genes in E9 and E10 WT and *Lyl-1^{LacZ/LacZ}* M ϕ -progenitors might raise more questions than provide answers. The importance of differentially expressed pathways would require the development of well-designed research projects.

Particularly interesting to investigate would be the inflammatory pathway, prevalent in the primitive population and dependent on *Lyl-1* expression, as the function of macrophages in the generation of Hematopoietic Stem cells, together with the importance of inflammatory signaling in this process is now well established.

3. In figure 6 (and suppl 4) the authors show a decreased rate of apoptosis and proliferation of Lyl-1-deficient microglia at E12, which might very well explain the similar frequency of microglial cells at E14, however, at E12, the population of Lyl-1-deficient microglia is nevertheless reduced (Fig 6a). Interestingly, Lyl-1-deficient microglia shows dramatic morphological changes at day E12, with deprivation of ramifications. Although the non-ramified microglia seem to preserve intact membrane (as of F4/80 staining) the authors should test whether those are late apoptotic cells non detectable by FACS measurement of AnnexinV (for example using TUNEL reaction and F4/80 staining).

Answer: This suggestion is a very good one: identifying apoptotic cells *in situ* in *CX3CR1^{WT/GFP}:Lyl-1^{LacZ/LacZ}* brain at E12 could have allowed to disclose information that would give more insight about *Lyl-1* function at this stage. It would have been particularly interesting to observe whether apoptotic cells are differentially distributed amongst various brain regions, or specifically located close to other brain cells type (neurons, vascular cells, astrocytes). Unfortunately, we could not perform these analyses as the mouse strain is no longer available to us.

4. The presence of these aberrant microglial cells is per se really interesting, and their functional implications should be further analyzed. Do these non-ramified microglial cells show defects in phagocytosis or neurons or environmental sampling, for example? Possibly in vitro approaches using neonatal-derived microglia could address this point.

Answer: In newborn, phagocytosis is recognized as important for synaptic pruning. We thus performed RT-qPCR analyses of genes involved in phagocytosis in WT and newborn microglia (*Axl*, *MertK*, *Trem2* and *P2Yr6*). We observed a significantly reduced expression of *Axl*, *MertK* and *P2Yr6* in mutant microglia, while *Trem2* expression was unmodified. These data now appear in figure 6J.

Interestingly, our RNA-seq. data showed that at E10, *Ly1-1^{LacZ/LacZ}* YS MΦ-progenitors expressed lower levels of these phagocytosis genes than WT. This reduction was only specific for *Axl*, due to the co-existence of primitive and transient-definitive MΦ-progenitors at this stage. These data now appear in Figure 4G

Reviewer #3: Production of selected immune cells, including macrophages, relies before the generation of HSC on two consecutive hematopoietic waves, primitive and transient definitive, that both arise in the YS and have also been termed 'early' (eEMP) and 'late' EMP. Most tissue macrophages, including microglia, are now appreciated to originate from these early waves, and subsequently self-maintain independent from HSC input. Differential contributions of eEMP and EMP to the macrophage compartments remain however less well defined.

Here the authors focus on the BLH transcription factor (TF) Lyl-1 (1) as marker for early primitive macrophage precursors in the YS, and (2) with respect to its function in microglia development.

In the first part the authors take advantage of published Lyl-1-lacZ reporter animals, in combination with the FDG assay, to establish that Lyl-1 expression allows to discriminate primitive from transient-definitive macrophage precursors in the YS. They then move on to compare differentiation potential of E9 and E10 MF progenitors, including short term YS organ cultures, flow cytometry, RNAseq analysis and clonogenic assays. The accumulation of MF precursors in the E8 Lyl-1 deficient YS suggests that in WT mice the TF restricts the progenitor pool size potentially by affecting the window of production.

The authors further report that the Lyl-1 deficiency results in a transient decrease of the microglia pool size. Since microglia themselves also express Lyle-1, it remains unclear whether this phenotype is due to affected primitive progenitors or direct effects on microglia.

Taken together, this is an interesting study addressing an important biological question. The conclusions were found largely supported by solid data, although the study is largely descriptive and statements are hence often based on correlations rather than formal proof of causality. The manuscript could be significantly improved in clarity and a clearer formulation of its key findings. At the moment, the reader is left with a number of interesting observations, but the main message remains somewhat elusive because elaborations are limited.

Specific points:

The authors seem to claim to provide evidence that microglia arise from early EMP. They base this conclusion on transcriptome overlap, including expression of the microglia signature gene Sall1. However, this part of the study, which would represent a major finding, came across as underdeveloped and could be carved out better.

Answer: While we think that our work points to a filiation between $M\Phi^{\text{Prim}}$ /early EMP and microglia development, we agree with the reviewer that it does not demonstrate it completely,

since our evidence is mostly based on transcriptome and not on a fate mapping marker. Therefore, we did not develop this point extensively as too much emphasis could be considered as an over interpretation by some reviewers.

1- The authors mention MHC II expression at E9 and E10 (line 169), addition of a representative FACS blot would here be convincing.

Answer: A representative cytometry profile of MHC-II expression in MΦ progenitors from E9 and E10 WT and *Lyl-1^{LacZ/LacZ}* YS was included in Supplemental figure 1D.

2-The authors note limited differentiation of E9 MF, and mention evidence in the sequencing data but this could be elaborated further. In line 238, the authors mention defective cytokine signaling (line 238), but it remains unclear how they got to that conclusion, Suppl. Fig 3D/E is here rather unclear.

Answer: In our initial submission, we chose not to develop this point in order to fit word number limitations. We now succinctly commented on the important modifications of Jak-Stat signaling pathway and of interleukin/interleukin receptors expression in E9 *Lyl-1^{LacZ/LacZ}* MΦ-progenitors.

3- The authors mention the identification of a *Lyl-1* core signature, but don't elaborate further.

Answer: The function of the 16 common DEGs on early stages of MΦ-progenitor development remains largely un-annotated. However, some (*Lyl-1*, *Asb2*, *Nfatc2*) were found to belong to microglia signatures¹, or involved in response to pathogens (*Lyl-1*, *Tbc1d9*)². Understanding the biological meaning of this core signature would require the development of full research projects together with suitable murine models. We found that commenting on the *Lyl-1* core signature would be highly speculative at this stage.

4- The contribution of *Lyl-1* to the BAM compartment could be expanded, by simply adding a FACS blot showing CD11b/ F4/80 or CD206/Cx3Cr1 (see Utz et al, ref 16). Are BAM affected in the *Lyl-1* KO?

Answer: Because the *Lyl-1^{LacZ/LacZ}* strain is no longer available to us, we reanalyzed our previous cytometry data in the attempt to gain more insight into *Lyl-1* expression and the impact of its deficiency on the BAM subset. In the absence of the markers that discriminate BAM from microglia (mostly CD206) according to Goldmann et al. and Utz et al.^{3, 4} in our previous cytometry analyses, we set out to identify BAM using the other phenotype differences described by these authors (F4/80^{Hi}CD11b⁺CD45⁺ for BAM and F4/80⁺CD11b^{Hi}CD45^{Lo} for Microglia). We assessed FDG/*Lyl-1* expression in BAM and compared the BAM population size in WT and *Lyl-1^{LacZ/LacZ}* mice at different development stages (E12, E14, P0-P3 and P42). The detailed results are shown below.

Briefly, while our gating strategy failed to clearly distinguish the BAM population from microglia, the BAM-enriched population appeared to express *Lyl-1* at all stages investigated, with a lower expression level in the adult. Moreover, the size of this population appeared increased at E14 and in the newborn. However, even if these expression data suggest that *Lyl-1* may play a role in BAM development, specific analyses involving a stringent separation of BAM from microglia in cytometry analyses, as well as imaging of this population in WT and *Lyl-1^{LacZ/LacZ}* brains, would be required to obtain conclusive information on *Lyl-1* function in the BAM population.

Detailed information: Tentative analysis of *Lyl-1* expression and function in BAMs

We set out to identify BAM, in the absence of the discriminating marker CD206, based on previously published cytometry, where embryonic BAMs were defined as CD45⁺F4/80^{High}CD11b^{Low}, while microglia were defined as CD45^{Low}F4/80⁺CD11b^{High}^{3, 4}.

This gating strategy (**Additional figure A**) failed to clearly separate the microglia and BAM subsets, but allowed a prospective characterization of a BAM-enriched population.

The analysis of FDG/*Lyl-1* expression in *Lyl-1^{WT/LacZ}* brain from E12 (**Additional figure B**) to P42 indicated that the BAM-enriched population expresses FDG/*Lyl-1* at all stages with levels slightly lower to similar to FDG/*Lyl-1* expression in microglia.

To evaluate possible defects of *Lyl-1* deficiency on the BAM-enriched population, we quantified both microglia and BAM-enriched population at E12, E14, P0-P3 and P42 (**Additional figure C**). The increase of the BAM-enriched subset at E12 and P0-P3 being possibly a consequence of the decreased size of the microglia population, we reported the percentage of Microglia and “BAM”

to the whole CD45+ population (**Additional figure D**). From this comparison, it appeared that the BAM-enriched subset might be increased in *Lyl-1^{LacZ/LacZ}* brains at E14 and in the newborn, but not at E12 and in the adult.

However, even if these expression data suggest that *Lyl-1* may play a role in BAM development, specific analyses involving a stringent separation of BAM from microglia in cytometry analyses, as well as imaging of this population in WT and *Lyl-1^{LacZ/LacZ}* brains, would be required to obtain conclusive information on *Lyl-1* function in the BAM population.

References

1. Bennett, M.L. *et al.* New tools for studying microglia in the mouse and human CNS. *Proceedings of the National Academy of Sciences* **113**, E1738-E1746 (2016).
2. Yan, Q. *et al.* Candidate genes on murine chromosome 8 are associated with susceptibility to *Staphylococcus aureus* infection in mice and are involved with *Staphylococcus aureus* septicemia in humans. *PLOS ONE* **12**, e0179033 (2017).
3. Goldmann, T. *et al.* Origin, fate and dynamics of macrophages at central nervous system interfaces. *Nature Immunology* **17**, 797-805 (2016).
4. Utz, S.G. *et al.* Early Fate Defines Microglia and Non-parenchymal Brain Macrophage Development. *Cell* **181**, 557-573 e518 (2020).

Additional figure

Figure legend

A- Gating strategy used to identify BAM: Top representative profiles, shown for E12 WT brain: Both microglia (Mgl) and BAM are included in the F4/80+CD45+ population.

Medium and lower profiles: according to ^{3, 4}, embryonic BAM are CD45⁺F4/80^{High}CD11b^{Low}, whereas microglia are CD45^{Low}F4/80⁺CD11b^{Low}.

B- FDG/Lyl-1 expression in E12 *Lyl-1*^{WT/LacZ} brain: The BAM-enriched subset (lower profile) expresses similar FDG/Lyl-1 to that of the microglia-enriched subset (upper profile). The contour plots in WT samples indicate the level of non-specific β -Gal activity. Representative profiles of 3 independent experiments at E12. Similar FDG/Lyl-1 expression levels were observed at E14 and P0-P3 in *Lyl-1*^{WT/LacZ} BAM-enriched subset. This expression level was lower in the adult (Data not shown). In *Lyl-1*^{LacZ/LacZ} BAM-enriched subset, FDG/Lyl-1 expression levels were consistently higher than in the *Lyl-1*^{WT/LacZ} subset (Data not shown).

C- Distribution of the Microglia and BAM-enriched subset in WT and *Lyl-1*^{LacZ/LacZ} brains within the CD11b⁺F4/80⁺ population. (E12: WT: n=6; *Lyl-1*^{LacZ/LacZ}: n=5 in 3 independent experiments; E14: WT: n=6; *Lyl-1*^{LacZ/LacZ}: n=7 in 4 independent experiments; P0: WT: n=9; *Lyl-1*^{LacZ/LacZ}: n=10 in 5 (WT) and 6 (*Lyl-1*^{LacZ/LacZ}) independent experiments; P42: WT: n=3; *Lyl-1*^{LacZ/LacZ}: n=6 in 3 independent experiments;); Error bars show mean \pm s.e.m.; Two tailed, unpaired Mann-Whitney *t*-test.

D- Distribution of the Microglia and BAM-enriched subset in WT and *Lyl-1*^{LacZ/LacZ} brains within the CD45⁺F4/80⁺ population. Left: Global effect of Lyl-1-deficiency on the CD45⁺F4/80⁺ brain population. Right: Frequency of microglia and BAM-enriched population in the CD45⁺F4/80⁺ population. (E12: WT: n=6; *Lyl-1*^{LacZ/LacZ}: n=5 in 3 independent experiments; P0: WT: n=9; *Lyl-1*^{LacZ/LacZ}: n=10 in 5 (WT) and 6 (*Lyl-1*^{LacZ/LacZ}) independent experiments); Error bars show mean \pm s.e.m.; Two tailed, unpaired Mann-Whitney *t*-test.

REVIEWERS' COMMENTS:

Reviewer #1 (Remarks to the Author):

Most concerns and comments were clearly addressed in the revised manuscript. One missing part is the experiment with Lyl-lacZ/lacZ mice. Overall, all the revision experiments strengthen the conclusion of this manuscript.

Reviewer #2 (Remarks to the Author):

The revised manuscript from Wang and collaborators is now complete and the authors have addressed all the questions that were arisen. We recommend the manuscript for publication after editing of some typos and correct formating (Figure 6 is duplicated).

Reviewer #3 (Remarks to the Author):

The authors have responded to my comments to my satisfaction. Together with their responses to the other reviewers comments te manuscript is significantly improved and I recommend to accept.